# Human immunodeficiency virus-1 induces host genomic R-loops and preferentially integrates its genome near the R-loop regions

Kiwon Park[1,2†], Dohoon Lee[3,4†], Jiseok Jeong[1,2], Sungwon Lee[1,2], Sun Kim[5], Kwangseog Ahn[1,2,6]*

[1]Center for RNA Research, Institute for Basic Science, Seoul, Republic of Korea; [2]School of Biological Sciences, Seoul National University, Seoul, Republic of Korea; [3]Bioinformatics Institute, Seoul National University, Seoul, Republic of Korea; [4]BK21 FOUR Intelligence Computing, Seoul National University, Seoul, Republic of Korea; [5]Department of Computer Science and Engineering, Seoul National University, Seoul, Republic of Korea; [6]SNU Institute for Virus Research, Seoul National University, Seoul, Republic of Korea

*For correspondence:
ksahn@snu.ac.kr

†These authors contributed equally to this work

Competing interest: The authors declare that no competing interests exist.

## eLife Assessment

This study presents two main findings regarding HIV-1 genomic integration. The first, based on **convincing** evidence in primary cell models, is that HIV-1 induces R loop formation, though the viral driver of this process remains undefined. The second, based on model cell systems with limited physiological relevance to HIV-1, is that a portion of HIV-1 genomes integrates in the vicinity of where R loops form. This finding has the potential to offer **fundamental** insight into HIV-1 integration, but the strength of the presented evidence was viewed as incomplete and needing additional validation by more direct experimental methods in order to understand what the mechanistic relationship between the formation of R loops and HIV-1 integration is.

**Abstract** Although HIV-1 integration sites favor active transcription units in the human genome, high-resolution analysis of individual HIV-1 integration sites has shown that the virus can integrate into a variety of host genomic locations, including non-genic regions. The invisible infection by HIV-1 integrating into non-genic regions, challenging the traditional understanding of HIV-1 integration site selection, is more problematic because they are selected for preservation in the host genome during prolonged antiretroviral therapies. Here, we showed that HIV-1 integrates its viral genome into the vicinity of R-loops, a genomic structure composed of DNA-RNA hybrids. VSV-G-pseudotyped HIV-1 infection initiates the formation of R-loops in both genic and non-genic regions of the host genome and preferentially integrates into R-loop-rich regions. Using a HeLa cell model that can independently control transcriptional activity and R-loop formation, we demonstrated that the exogenous formation of R-loops directs HIV-1 integration-targeting sites. We also found that HIV-1 integrase proteins physically bind to the host genomic R-loops. These findings provide novel insights into the mechanisms underlying retroviral integration and the new strategies for antiretroviral therapy against HIV-1 latent infection.

## Introduction

Retroviruses cause permanent host infections by integrating their reverse-transcribed viral genomes into the host genome. Retroviral integration considerably affects a wide range of biological phenomena, including the persistence of fatal human diseases and the shaping of metazoan evolution (*Johnson, 2019*). Human immunodeficiency virus (HIV)–1 is a representative retrovirus underlying the global burden of acquired immune deficiency syndrome (AIDS) (*Lusic and Siliciano, 2017*). The chromosomal landscape of HIV-1 integration has been shown to influence proviral gene expression, the persistence of integrated proviruses, and prognosis in antiretroviral therapy (; *Einkauf et al., 2022*; *Jiang et al., 2020*). Integration into the host genome is not random and displays distinct preferences for gene-dense regions, where active transcription occurs (*Schröder et al., 2002*), by interacting with host factors, such as transcription activators, epigenetic marker binding proteins, and super-enhancers (*Achuthan et al., 2018*; *Ciuffi et al., 2005*; *Sowd et al., 2016*; *Lucic et al., 2019*; *Marini et al., 2015*; *Kvaratskhelia et al., 2014*; *Cherepanov et al., 2003*). However, transcriptional activity is not the sole determinant of the HIV-1 integration site landscape (*Lucic et al., 2019*). For instance, some highly expressed genes do not necessarily correspond to a high level of integration in PBMCs infected in cultures with HIV (*Coffin et al., 2021*). Despite their lower probability of integration, HIV-1 proviruses are observed in non-genic regions of the genomes of infected individuals (*Einkauf et al., 2022*). This indicates the possibility of there being an undiscovered mechanism or determinant that constitutes the correct genomic environment for HIV-1 integration.

An R-loop is a three-stranded nucleic acid structure that comprises a DNA-RNA hybrid and a displaced strand of DNA, and has long been considered a transcription byproduct (*Niehrs and Luke, 2020*; *Petermann et al., 2022*). R-loops in cellular genomes are enriched in actively transcribed genes as they occur naturally during transcription (*Niehrs and Luke, 2020*; *Hamperl et al., 2017*). However, R-loop formation is not limited to gene body regions and is widespread in the genome (*Niehrs and Luke, 2020*). As a result of in trans-R-loop formation, R-loops are also abundant in non-genic regions, such as intergenic regions and repetitive sequences, including transposable elements, centromeres, and telomeres (*Niehrs and Luke, 2020*; *Ginno et al., 2012*; *Lim et al., 2015*; *Arora et al., 2014*), independent of the transcriptional activity of the genes harboring the R-loops. Although R-loops have been identified as critical intermediates and regulators of a number of biological processes (*Niehrs and Luke, 2020*; *Petermann et al., 2022*; *García-Muse and Aguilera, 2019*), the dynamics and the role played by cellular R-loops in pathological contexts remain unclear.

R-loops are important contributors to molding the genomic environment and spatial organization of the cellular genome, and can potentially play a novel role in host-pathogen interaction. In the cellular genome, R-loops relieve superhelical stresses and are often associated with open chromatin marks and active enhancers (*Sanz et al., 2016*; *Chédin, 2016*), which are also distributed over HIV-1 integration sites (*Schröder et al., 2002*; *Sowd et al., 2016*; *Lucic et al., 2019*). In transcription-induced R-loop formation, a guanine-quadruplex (G4) structure can be generated in the non-template DNA strand of the R-loop (*Lee et al., 2020*). A recent study showed that G4 DNA can influence both productive and latent HIV-1 integration (*Ajoge et al., 2022*). In addition, R-loops are prevalent non-canonical B-form DNA structures (*Chedin and Benham, 2020*) and intermediates between the B-form DNA and A-form RNA conformations (*Jóźwik et al., 2022*). It has recently been reported that the B-to-A transition in the target DNA occurs during retroviral integration (*Jóźwik et al., 2022*; *Ballandras-Colas et al., 2022*). Accumulating evidence suggests that host genomic R-loops are undiscovered host factors in the HIV-1 integration site selection mechanism that dynamically interact with the host genomic environment.

Here, we showed a notable role for R-loops in the interaction between HIV-1 and its host, specifically in HIV-1 integration. VSV-G-pseudotyped HIV-1 infection induces host cellular R-loop formation, and the R-loop-rich regions of the host genome are preferred for HIV-1 integration in diverse cell types. HIV-1 integrase proteins showed considerable binding affinity for nucleic acid substrates containing R-loop structures. Our results suggest that R-loops are important components of the host genomic environment for HIV-1 integration site determination.

## Results

### Host genomic R-loops accumulate by VSV-G-pseudotyped HIV-infection in diverse cell types

To investigate the relationship between HIV-1 infection and host cellular R-loops, we first analyzed R-loop dynamics in different types of cells infected with HIV-1 at early post-infection time points using DNA-RNA immunoprecipitation followed by cDNA conversion coupled with high-throughput sequencing (DRIPc-seq) using a DNA-RNA hybrid-specific binding antibody, anti-S9.6 (*Sanz and Chédin, 2019*). HeLa cells, primary CD4$^+$ T cells isolated from two individual donors, and the CD4$^+$/CD8$^-$ T cell lymphoma Jurkat cell line were infected with the VSV-G-pseudotyped HIV-1-EGFP in a time-course manner. The infected cells were harvested at the same post-seeding time point but at 0, 3, 6, and 12 hr post-infection (hpi) for DRIPc-seq library construction (*Figure 1A* and *Figure 1—figure supplement 1A–C*). Our DRIPc-seq analysis yielded loci-specific R-loop signals at the referenced R-loop-positive loci (RPL13A and CALM3) and an R-loop-negative locus (SNRPN) (*Sanz and Chédin, 2019*), which were both strand-specific and highly sensitive to pre-immunoprecipitation in vitro RNase H treatment in HeLa cells, CD4$^+$, and Jurkat T cells (*Figure 1—figure supplements 2–4*). Notably, the number of DRIPc-seq peaks mapped to the human reference genome increased markedly during the early post-infection with HIV-1 (3 and 6 hpi for HeLa cells and 6 and 12 hpi for CD4$^+$ and Jurkat T cells; *Figure 1B*). Most of the peaks that were mapped in cells harvested at 0 hpi were commonly found in all other samples, but a significant number of unique peaks were observed after infection (*Figure 1C*).

In addition to the DRIPc-seq data analysis, we used different biochemical approaches to examine R-loop accumulation after HIV-1 infection in HeLa cells. R-loop accumulation in HIV-1-infected cells was observed using DNA-RNA hybrid dot blots with the anti-S9.6 antibodies (*Figure 1D*). Dot intensity significantly increased upon HIV-1 infection at 6 hpi and the enhanced R-loop signals on the dot blots of HIV-1-infected cells were highly sensitive to in vitro treatment with RNase H (*Figure 1D*). This result is highly consistent with our DRIPc-seq data analysis of HIV-1-infected HeLa cells. Subsequently, we observed HIV-1-induced R-loops using an immunofluorescence assay by probing HIV-1-infected or non-infected control cells with S9.6 antibody at 6 hpi (*Figure 1E*). After subtracting the nucleolar signal, the nuclear fluorescence signal associated with R-loops was significantly enhanced in cells infected with HIV-1 (*Figure 1E*). We validated and quantified HIV-1-infection-induced R-loop formation on the host genome in a genome-site-specific manner by using DRIP followed by real-time polymerase chain reaction (DRIP-qPCR). In this experiment, the S9.6 signal was determined for three and two HIV-1-induced-R-loop-positive (P1, P2, and P3) and -negative regions (N1 and N2), respectively, which were defined by DRIPc-seq data analysis (*Figure 1—figure supplement 5A–E*). We detected significantly increased R-loop signals that were highly sensitive to RNase H treatment of pre-immunoprecipitates in the P1, P2, and P3 regions of HIV-1-infected cells at 6 hpi compared to those in the cells harvested at 0 hpi (*Figure 1—figure supplement 6A*). However, the HIV-1-induced R-loop-negative regions N1 and N2, did not show significant R-loop accumulations (*Figure 1—figure supplement 6A*).

Importantly, the R-loop signal was enriched even in cells infected with HIV-1 when the reverse transcription or integration of HIV-1 was blocked by enzyme inhibitors, such as nevirapine (NVP) or raltegravir (RAL) (*Figure 1—figure supplement 6B and C*). This result indicates that the enrichment of R-loop signals in cells originates from the host genome, but not from DNA-RNA hybrid formation during the viral life cycle or transcriptional burst from integrated HIV-1 proviruses. In addition, we confirmed that nearly 100% of DRIPc-seq reads from HIV-1-infected HeLa, CD4$^+$, and Jurkat T cells were aligned to the host cellular genome, but not to that of HIV-1 (*Figure 1—figure supplement 6D*). Together, these data demonstrated that HIV-1 infection induces host genomic R-loop enrichment at early post-infection. This also suggests that HIV-1 infection regulates R-loop dynamics in host cells, possibly by inducing either new R-loop formation or stabilizing existing R-loops in the host cellular genome.

### R-loops accumulation after HIV-1 infection are widely distributed in both genic and non-genic regions

To investigate the distribution of cellular genomic R-loops during HIV-1 infection, we conducted a genome-wide analysis of the DRIPc-seq data. Unique DRIPc-seq peaks observed after HIV-1-infection

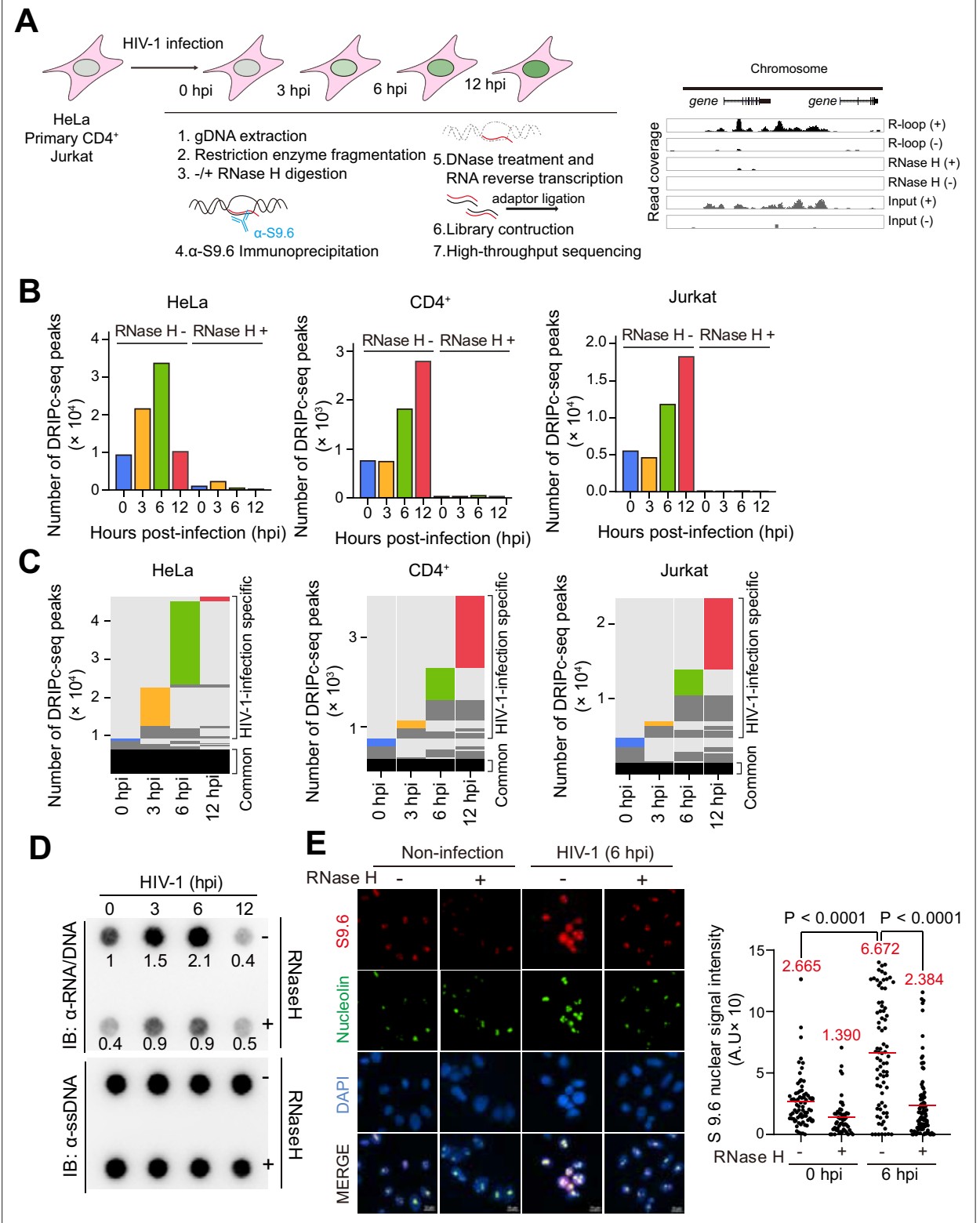

**Figure 1.** HIV-1 infection induces genomic R-loop accumulation in cells at early post-infection. (**A**) Summary of experimental design for DNA-RNA immunoprecipitation followed by cDNA conversion coupled with high-throughput sequencing (DRIPc-seq) in HeLa cells, primary CD4+ T cells, and Jurkat cells infected with HIV-1. (**B**) Bar graphs indicating DRIPc-seq peak counts for HIV-1-infected HeLa cells, primary CD4+ T cells, and Jurkat cells harvested at the indicated hours post-infection (hpi). Pre-immunoprecipitated samples were untreated (−) or treated (+) with RNase H, as indicated. Each bar corresponds to pooled datasets from two biologically independent experiments. (**C**) All genomic loci overlapping a DRIPc-seq peak from HIV-1

*Figure 1 continued on next page*

*Figure 1 continued*

infected HeLa cells, primary CD4$^+$ T cells, and Jurkat cells in at least one sample are stacked vertically; the position of each peak in a stack is constant horizontally across samples. Each hpi occupies a vertical bar, as indicated. Each bar corresponds to pooled datasets from two biologically independent experiments. Common peaks for all samples are represented in black, and in dark gray for those common for at least two samples. The lack of a DRIP signal over a given peak in any sample is shown in light gray. The sample-unique peaks are colored blue, yellow, green, and red at 0, 3, 6, and 12 hpi, respectively. (**D**) Dot blot analysis of the R-loop in gDNA extracts from HIV-1 infected HeLa cells with multiplicity of infection (MOI) of 0.6 harvested at the indicated hpi. gDNAs were probed with anti-S9.6. gDNA extracts were incubated with or without RNase H in vitro before membrane loading (anti-RNA/DNA signal). Fold-induction was normalized to the value of harvested cells at 0 hpi by quantifying the dot intensity of the blots and calculating the ratios of the S9.6 signal to the total amount of gDNA (anti-ssDNA signal). (**E**) Representative images of the immunofluorescence assay of S9.6 nuclear signals in HIV-1 infected HeLa cells with MOI of 0.6 harvested at 6 hpi. The cells were pre-extracted from cytoplasm and co-stained with anti-S9.6 (red), anti-nucleolin antibodies (green), and DAPI (blue). The cells were incubated with or without RNase H in vitro before staining with anti-S9.6 antibodies, as indicated. Quantification of S9.6 signal intensity per nucleus after nucleolar signal subtraction for the immunofluorescence assay. The mean value for each data point is indicated by the red line. Statistical significance was assessed using one-way ANOVA (n >53).

The online version of this article includes the following source data and figure supplement(s) for figure 1:

**Source data 1.** Original dot blots for *Figure 1D*, indicate the relevant dots and treatments.

**Source data 2.** Original files for dot blot analysis are displayed in *Figure 1D*.

**Figure supplement 1.** Primary CD4$^+$ T cells sorting strategies and GFP-HIV-1 infection.

**Figure supplement 2.** Chromosomal position and DNA-RNA immunoprecipitation followed by cDNA conversion coupled with high-throughput sequencing (DRIPc-seq) signal for referenced R-loop-positive and –negative regions in HIV-1 infected HeLa cells.

**Figure supplement 2—source data 1.**

**Figure supplement 3.** Chromosomal position and DNA-RNA immunoprecipitation followed by cDNA conversion coupled with high-throughput sequencing (DRIPc-seq) signal for referenced R-loop-positive and –negative regions in HIV-1 infected primary CD4$^+$ T cells.

**Figure supplement 3—source data 1.**

**Figure supplement 4.** Chromosomal position and DNA-RNA immunoprecipitation followed by cDNA conversion coupled with high-throughput sequencing (DRIPc-seq) signal for referenced R-loop-positive and –negative regions in HIV-1 infected Jurkat cells.

**Figure supplement 4—source data 1.**

**Figure supplement 5.** Genome browser screenshot over the HIV-1-induced R-loop forming positive or negative genomic regions.

**Figure supplement 6.** Host cellular R-loop induction by HIV-1 infection is host-genome specific.

**Figure supplement 6—source data 1.** Original dot blots for *Figure 1—figure supplement 6B*, indicating the relevant dots and treatments.

**Figure supplement 6—source data 2.** Original files for dot blot analysis are displayed in *Figure 1—figure supplement 6B*.

were numerous and relatively long (*Figure 2A*). This suggests that the R-loops induced by HIV-1 infection occupy a genomic region larger than that of the R-loops without HIV-1 infection. We observed a significant accumulation of R-loops over diverse genomic compartments at the hpi of HIV-1-infection induced R-loop formation (*Figure 2B*). The presence of R-loops is often correlated with high transcriptional activity, and we found a significantly high proportion of DRIPc-seq peaks enrichment upon HIV-1 infection in the gene body regions (*Figure 2B*). However, we also observed enrichment of HIV-1-infection-induced DRIPc-seq peaks proportions mapped to intergenic or repeat regions, including short interspersed nuclear elements (SINEs), long interspersed nuclear elements (LINEs), and long terminal repeat (LTR) retrotransposons, where transcription is typically repressed (*Figure 2B*). In addition, the proportion of DRIPc-seq peaks mapped to various genomic compartments remained consistent over the hours following HIV-1 infection (*Figure 2C*). This suggests that HIV-1 infection does not induce R-loop enrichment at specific genomic features, but that R-loop accumulation after HIV-1 infection is widely distributed. Although the expression of repetitive elements is mostly repressed during normal cellular activities, HIV-1 infection could activate endogenous retroviral promoters (*Jones et al., 2013*; *Srinivasachar Badarinarayan et al., 2020*). To investigate whether R-loop induction in gene-silent regions is associated with transcriptome changes during HIV-1 infection, we performed RNA sequencing (RNA-seq) of HIV-1-infected HeLa cells at 0, 3, 6, and 12 hpi. Consistent with previous reports, we observed an increase in the expression levels of repetitive elements at later time points post-infection (*Figure 2—figure supplement 1A*). In contrast, we found that the expression levels of SINEs, LINEs, and LTRs were even lower at both 3 and 6 hpi compared with those at 0 hpi whereas HIV-1-induced R-loops were significantly accumulated compared with those at 0 hpi (*Figure 2—figure supplement 1A*). We further examined the expression profiles of genes containing the R-loop in HeLa cells. The expression profile of genes harboring HIV-1-induced R-loops in their

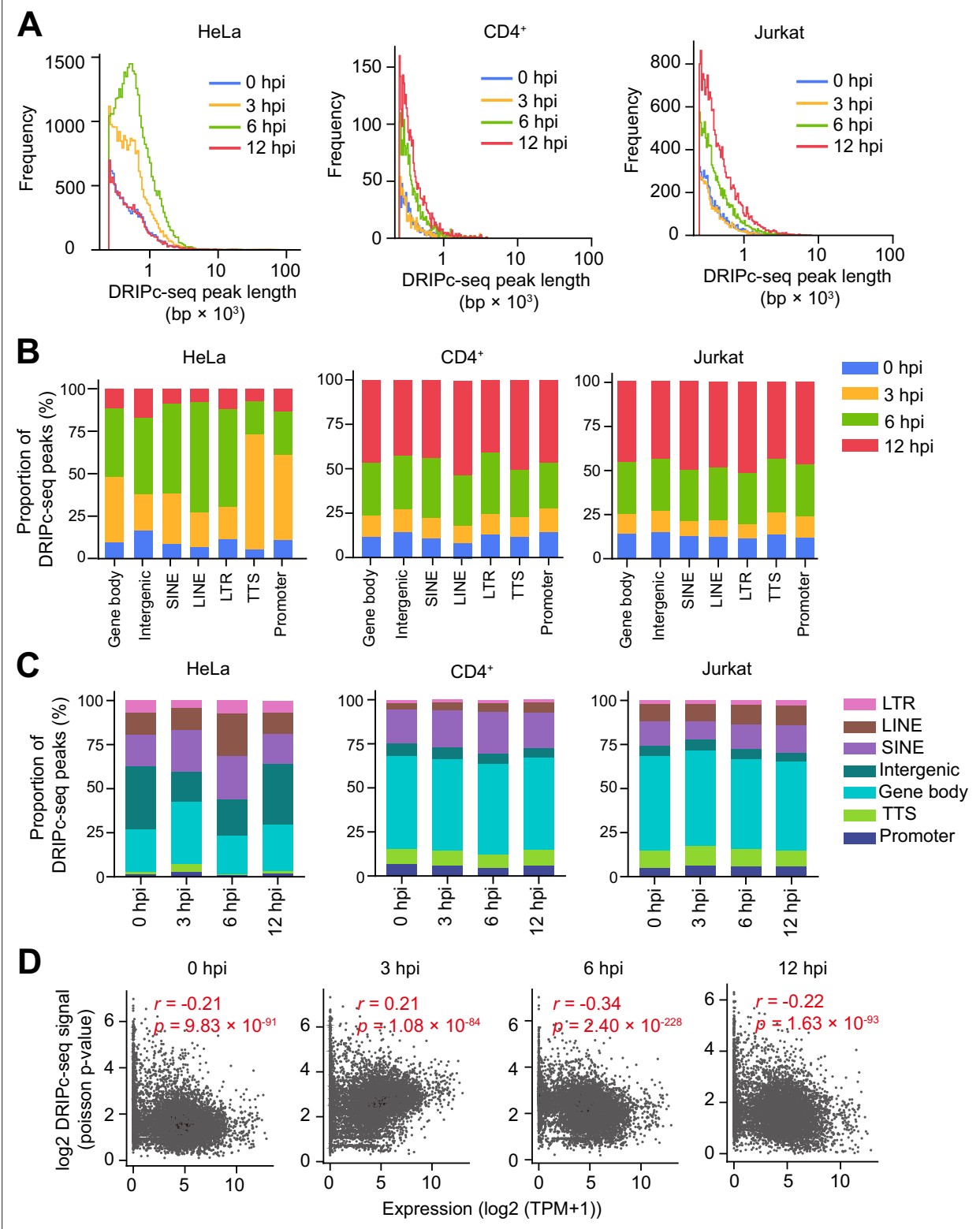

**Figure 2.** HIV-1-induced R-loops are enriched at both transcriptionally active and silent regions. (**A**) Distribution of DNA-RNA immunoprecipitation followed by cDNA conversion coupled with high-throughput sequencing (DRIPc-seq) peak lengths for HIV-1-infected HeLa cells, primary CD4+ T cells, and Jurkat cells harvested at the indicated time points (blue, 0 hpi; yellow, 3 hpi; green, 6 hpi; red, 12 hpi). (**B**) Stacked bar graphs indicating the proportion of DRIPc-seq peaks mapped for HIV-1-infected HeLa cells, primary CD4+ T cells, and Jurkat cells harvested at the indicated hpi over different genomic features. (**C**) Stacked bar graphs indicating the proportion of DRIPc-seq peaks mapped to indicated genomic compartments for HIV-1-infected

*Figure 2 continued on next page*

*Figure 2 continued*

HeLa cells, primary CD4$^+$ T cells and Jurkat cells harvested at the 0, 3, 6, and 12 hpi. (**D**) Correlation between gene expression and DRIPc-seq signals of HIV-1-infected HeLa cells with multiplicity of infection (MOI) of 0.6 harvested at the indicated hpi. Statistical significance was assessed using Pearson's r and p-values.

The online version of this article includes the following source data and figure supplement(s) for figure 2:

**Figure supplement 1.** R-loop induction by HIV-1 infection does not follow transcriptome changes in HeLa cells.

**Figure supplement 2.** RNA-seq analysis of relative gene expression levels of P1-3 and N1,2 R-loop regions.

**Figure supplement 2—source data 1.**

gene bodies showed very weak correlations with the signals of DRIPc-seq peaks at 3 hpi (Pearson's r=0.21, p=1.08 × 10$^{-84}$; *Figure 2D*) and at 6 hpi (Pearson's r=–0.34, p=2.40 × 10$^{-228}$; *Figure 2D*), which implies that the unique R-loop peaks upon HIV-1 infection do not engage in a transcriptional burst. In agreement with our DRIPc-seq and global RNA-seq data analysis, the expression level of the genes harboring HIV-1-infection-induced R-loops, which were quantified by DRIP-qPCR (*Figure 1—figure supplement 6A*), were not significantly affected by HIV-1 infection (*Figure 2—figure supplement 1B* and *Figure 2—figure supplement 2*). Together, our data demonstrate that host cellular R-loop accumulation upon HIV-1 infection is widely distributed in both genic and non-genic regions and is not necessarily correlated with the expression levels of the genes harboring the R-loops.

## HIV-1 integration sites are enriched at systemically induced sequence-specific R-loop regions in a cell model

HIV-1 infection is completed by integrating the viral genome into the hosts through dynamic inter-actions with the host genome (*Lesbats et al., 2016*). In addition, as HIV-1 infection induces R-loop accumulation at early post-infection hours, when the HIV-1 genome is imported into the nucleus and integration may initiate (*Brussel and Sonigo, 2003*; *Albanese et al., 2008*; *Dharan et al., 2020*), we hypothesized that host genomic R-loops play a role in HIV-1 integration and possibly in integration site selection. To systemically and directly assess the relationship between host genomic R-loops and HIV-1 integration in a genome site-specific manner, we adapted and modified an elegantly designed episomal system that induces sequence-specific R-loops through DOX-inducible promoters (*Hamperl et al., 2017*). To most closely mimic the presence of the R-loop in the host cellular genome, we subcloned the R-loop-forming portion of the mouse gene encoding AIRN (mAIRN) (*Ginno et al., 2012*) or the non-R-loop-forming ECFP sequence (*Hamperl et al., 2017*) with a DOX-inducible promoter into the piggyBac transposon vector and co-expressed piggyBac transposase in HeLa cells. The mAIRN gene sequence possesses a high GC skew that causes R-loop structure formation upon transcription by hybridizing newly synthesized RNA back to the template DNA strand, and the non-template DNA strand to remains looped out in a single-stranded form (*Ginno et al., 2012*; *Ginno et al., 2013*). In contrast, the ECFP sequence possesses a low GC skew that does not form a stable R-loop at the site of transcription. Thus, it was used as an R-loop-forming negative sequence. These R-loop forming (mAIRN) or non-R-loop forming (ECFP) sequences are nonhuman sequences. Therefore, our cell model allowed us to induce and quantify R-loop formation at designated genomic regions and distinguish R-loop formation from endogenous R-loops on the cellular genome, which are not sequence-specific and impossible to control for induction. Moreover, using this system we can quantify R-loop-associated site-specific HIV-1 integration events at designated regions, which can also be distinguished from HIV-1 integration events at endogenous host genomic loci. We designated the pool of cells with the R-loop forming sequence (mAIRN) inserted into its genome as a 'pgR-rich (piggyBac R-loop rich)' cell line and the pool of cells with the non-R-loop forming sequence (ECFP) inserted into its genome as 'pgR-poor (piggyBac R-loop poor)' cell line (*Figure 3A*).

A similar number of the copies of the piggyBac transposon were successfully delivered to the genome of each cell line (*Figure 3—figure supplement 1A*) and DOX treatment strongly induced the transcriptional activity of mAIRN or ECFP without affecting the transcription of endogenous loci in either cell line (*Figure 3—figure supplement 1B and C*). Although the transcription of mAIRN or ECFP was strongly induced by DOX treatment, the activity did not exceed that of the endogenous loci in either cell line (*Figure 3—figure supplement 1D and E*). Although the two cell lines showed comparable levels of DOX-inducible transcriptional activity at the designated sequences (*Figure 3B*),

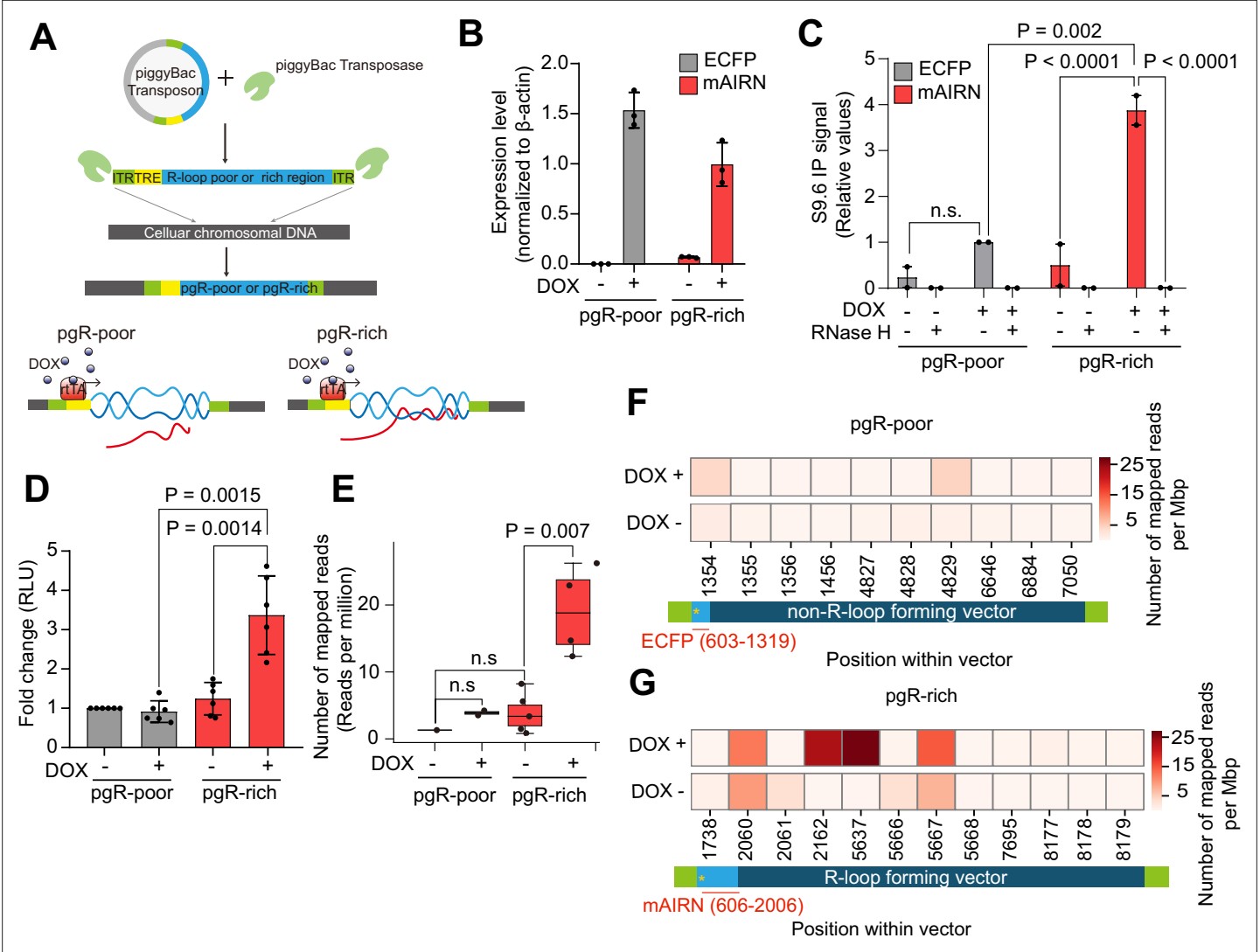

**Figure 3.** R-loop inducible cell line model directly addresses R-loop-mediated HIV-1 integration site selection. (**A**) Summary of the experimental design for R-loop inducible cell lines, pgR-poor and pgR-rich. (**B**) Gene expression of ECFP (gray) and mAIRN (red), as measured using RT-qPCR in pgR-poor or pgR-rich cells. Where indicated, the cells were incubated with 1 µg/ml DOX for 24 hr. Gene expression was normalized relative to *β-actin*. Data are presented as the mean ± SEM, n=3. (**C**) DRIP-qPCR using the anti-S9.6 antibody against ECFP and mAIRN in pgR-poor or pgR-rich cells. Where indicated, the cells were incubated with 1 µg/ml DOX for 24 hr. Pre-immunoprecipitated samples were untreated or treated with RNase H as indicated. Values are relative to those of DOX-treated (+) RNase H-untreated (−) pgR-poor cells. Data are presented as the mean ± SEM; statistical significance was assessed using two-way ANOVA (n = 2). (**D**) Bar graphs indicate luciferase activity at 48 hpi in pgR-poor or prR-rich cells infected with 100 ng/p24 capsid antigen of luciferase reporter HIV-1 virus per $1 \times 10^5$ cells/mL. Data are presented as the mean ± SEM; p-values were calculated using one-way ANOVA (n=6). (**E**) Box graph indicating the quantified HIV-1 integration site sequencing read count across pgR-poor and pgR-rich transposon sequences in untreated (–) or DOX-treated (+) pgR-poor or pgR-rich cell line infected with 100 ng/p24 capsid antigen of luciferase reporter HIV-1 virus per $1 \times 10^5$ cells/mL. Each bar corresponds to pooled datasets from three biologically independent experiments (n=3). In each boxplot, the centerline denotes the median, the upper and lower box limits denote the upper and lower quartiles, and the whiskers denote the 1.5x interquartile range. Statistical significance was assessed using a two-sided Mann–Whitney U test. (**F and G**) Heat maps representing the number of HIV-1 integration-seq mapped read across pgR-poor (**F**) or pgR-rich (**G**) transposon sequence in untreated (-) or DOX-treated (+) pgR-poor (**F**) or pgR-rich (**G**) cell line. Each rectangular box corresponds to the pooled the number of HIV-1 integration-seq mapped read from three biologically independent experiments (n=3) at the indicated position within pgR-poor (**F**) or pgR-rich (**G**) transposon vector. Each light blue box represents the actual position of the R-loop forming or non-R-loop forming sequence (ECFP or mAIRN) and the yellow stars indicate the TRE promoter position within a vector.

The online version of this article includes the following figure supplement(s) for figure 3:

**Figure supplement 1.** PiggyBac transposon-transposase insertion of R-loop forming and non-R-loop forming sequences in HeLa cells.

only the pgR-rich cells exhibited robust RNase H-sensitive stable R-loop formation upon DOX treatment (*Figure 3C*, mAIRN). In contrast, R-loops were weakly formed in the pgR-poor cells where a non-R-loop-forming sequence (ECFP) was inserted into the genome (*Figure 3C*, ECFP).

To examine whether the formation of 'extra' R-loops in the host genome influences HIV-1 infection in host cells, we infected both cell lines with VSV-G-pseudotyped HIV-1-luciferase viruses and examined the luciferase activity. Notably, we found that pgR-rich cells showed significantly higher luciferase activity only when R-loops were induced by DOX treatment, whereas pgR-poor cells showed comparable luciferase activity regardless of transcriptional activation by DOX treatment (*Figure 3D*). We sequenced HIV-1 integration sites in HIV-1-infected pgR-poor and pgR-rich cells to directly quantify site-specific integration events in sequence-specific R-loop regions. Remarkably, integration events were significantly higher in pgR-rich cells only when R-loops were induced by DOX treatment (*Figure 3E*). However, the number of HIV-1 integration-seq mapped reads within the non-R-loop forming sequence in pgR-poor cells remained very low, even after transcriptional activation by DOX treatment (*Figure 3E*). HIV-1 integration was enriched in the vicinity of R-loop forming regions in the pgR-rich cell line upon DOX treatment, but the enrichment was not observed in pgR-poor cells that did not form stable R-loops, even after transcriptional activation by DOX treatment (*Figure 3F and G*). This cell-based R-loop-inducing system with independent control over transcription and R-loop formation enabled the direct measurement of HIV-1 integration events at defined R-loop regions. The results indicated that host genomic R-loops are preferred by HIV-1 integration. Moreover, our data suggest that transcriptional activity itself is not sufficient for HIV-1 integration site determination, but that the formation of R-loops accounts for HIV-1 integration site selection.

## Host genomic R-loop regions are frequently targeted by HIV-1 integration

We attempted to further validate the relationship between R-loops and HIV-1 integration site selection by global analysis of HIV-1 integration sites in the endogenous genomic regions of HIV-1 infected host cells. We performed HIV-1 integration site sequencing in HIV-1 infected HeLa cells, CD4[+], and Jurkat T cells, and analyzed the sequencing data combined with our DRIPc-seq data. We counted and compared the number of successfully integrated proviruses in the R-loop regions (the combined genomic regions within 30 kb windows centered on DRIPc-seq peaks from 0, 3, 6, and 12 hpi) to those in non-R-loop-forming regions (the total genomic regions outside of the 30 kb windows centered on DRIPc-seq peaks). Notably, we detected approximately three to four times more integration in the R-loop regions than in other genomic regions without R-loops in HeLa cells, CD4[+], and Jurkat T cells (*Figure 4A*). Notably, HIV-1 integration sites preferred the centeral and nearby areas of the R-loops regions (*Figure 4B*). We observed biases in HIV-1 integration in HIV-1-induced R-loop-positive regions, P1-P3, which showed highly induced R-loop signals upon HIV-1 infection in DRIPc-seq analysis and DRIP-qPCR (*Figure 4C*). In contrast, HIV-1 integration sites were not detected in the R-loop-negative regions N1 and N2 (*Figure 4D*). Overall, our results from bioinformatics analysis using different types of naïve host cells infected with HIV-1 are consistent with the idea that the virus has a preference for targeting R-loop-forming regions for integration (*Figure 3*), suggesting that R-loops are an important component of the host genomic environment for HIV-1 integration site determination.

## HIV-1 integrase physically interacts with R-loops on the host genome

The HIV-1 intasome is tethered to the host genome for viral cDNA integration. Intasomes consist of HIV-1 viral cDNA and the HIV-1 integrase proteins. We observed that HIV-1 was preferentially integrated into the R-loops regions in the host genome; thus, we hypothesized that the HIV-1 integrase protein could directly bind and be recruited to the genomic R-loops. To test this hypothesis, we investigated whether HIV-1 integrase proteins have a physical binding affinity for nucleic acid substrates with an R-loop structure. Although HIV-1 integrases are DNA and RNA binding proteins (*Kessl et al., 2016*; *van Gent et al., 1991*), their ability to bind to a three-stranded nucleic acid structure that is composed of a DNA–RNA hybrid-like R-loop has not been investigated. We carried out an in vitro protein-nucleic acid binding assay by electrophoretic mobility shift assay (EMSA) with Sso7d-tagged HIV-1 integrase (E152Q) recombinant proteins and diverse structures of nucleic acid substrates including R-loops and simple dsDNA duplexes. In this experiment, we used HIV-1 integrase protein with an active site amino acid substitution, E152Q, to prevent any undesirable alteration in nucleic acid substrates by enzymatic

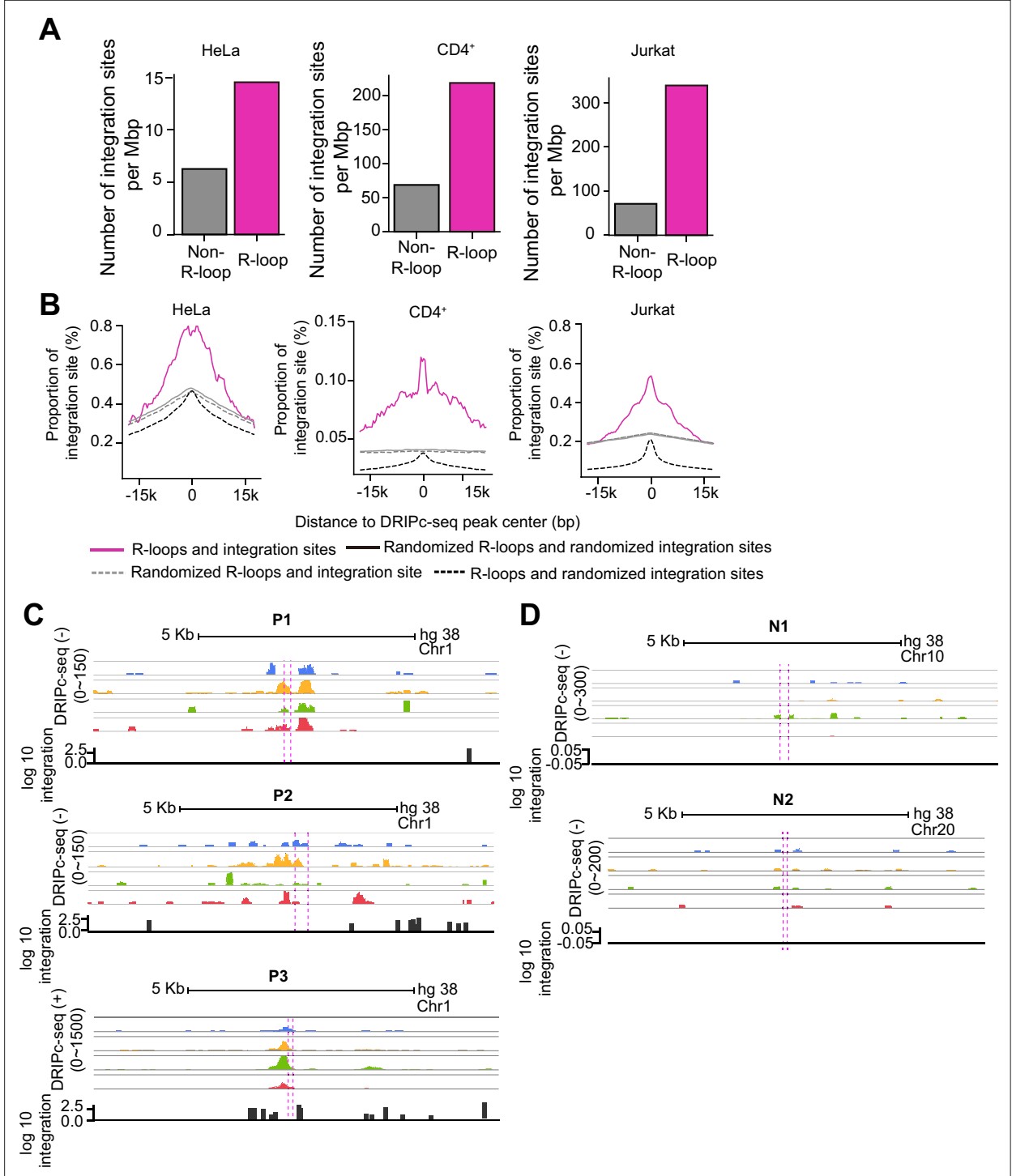

**Figure 4.** HIV-1 prefers host genomic R-loop regions for its viral cDNA integration. (**A**) Bar graphs showing the quantified number of HIV-1 integration sites per Mb pairs in total regions of 30 kb windows centered on DNA-RNA immunoprecipitation followed by cDNA conversion coupled with high-throughput sequencing (DRIPc-seq) peaks from HIV-1 infected HeLa cells, primary CD4+ T cells, and Jurkat cells (magenta) or non-R-loop region in the cellular genome (gray). (**B**) Proportion of integration sites within the 30-kb windows centered on DRIPc-seq peaks (magenta solid lines) or randomized DRIPc-seq peaks (gray dotted lines). Control comparisons between randomized integration sites with DRIPc-seq peaks and randomized DRIPc-seq peaks are indicated by black dotted lines and gray solid lines, respectively. (**C and D**) Superimpositions of HIV-1-induced R-loop positive chromatin regions, P1-P3 (**C**), and HIV-1-induced R-loop negative chromatin regions, N1 and N2 (**D**), on DRIPc-seq (blue, 0 hpi; yellow, 3 hpi; green, 6 hpi; red, 12 hpi) and the number of mapped read of HIV-1 integration-seq (integration, black). Magenta dotted lines represent primer binding sites in qPCR following DRIP.

activities of integrase proteins, such as 3'-processing. Notably, the nucleic acid substrate consisted of an R-loop structure bound to HIV-1 integrase proteins with higher binding affinity than that of the simple dsDNA duplex (*Figure 5A*). Additionally, the R-loop composing forms of nucleic acid structures, such as RNA-DNA hybrid with exposed ssDNA (R:D+ssDNA) and RNA-DNA hybrid (hybrid), also showed high binding affinity to integrases (*Figure 5—figure supplement 1A* and *Figure 5A*).

We validated the interaction between cellular genomic R-loops and HIV-1 integrase proteins by DNA-RNA hybrid immunoprecipitation using S9.6 antibodies against FLAG-tagged HIV-1 integrase-expressing HeLa cells (*Figure 5B*). Under our experimental conditions, R-loops were reproducibly immunoprecipitated (*Figure 5—figure supplement 1B*) and HIV-1 integrase proteins co-immuno-precipitated with the R-loops (*Figure 5C*). DNA-RNA hybrids also co-immunoprecipitated with the positive control H3 (*Cristini et al., 2018*) but not with the negative controls LaminA/C and Actin (*Cristini et al., 2018*; *Figure 5C*). To verify the specificity of our co-immunoprecipitation results for R-loops and HIV-1 integrases, we performed DNA-RNA hybrid immunoprecipitation with RNase H treatment (*Figure 5—figure supplement 1B, C*). The S9.6 signal of the immunoprecipitated nucleic acids were highly sensitive to RNase H treatment of the pre-immunoprecipitates (*Figure 5D*). Accordingly, the blotting signal of the co-immunoprecipitated HIV-1 integrase and H3 proteins was significantly reduced upon RNase H treatment (*Figure 5E*). We performed reciprocal immunoprecipitation using an anti-FLAG monoclonal antibody and detected the immunoprecipitated R-loops using dot blot analysis with anti-S9.6. R-loops, which were immunoprecipitated using HIV-1 integrase, and the S9.6 signal of immunoprecipitated nucleic acids was highly sensitive to RNase H treatment (*Figure 5F* and *Figure 5—figure supplement 1D*). Subsequently, we examined the interaction between the R-loops and HIV-1 integrase using a proximity-ligation assay (PLA) in HIV-1-infected cells. We used two antibodies: one that binds to the R-loops (anti-S9.6) and another that binds to GFP-tagged HIV-1 integrase. We detected PLA signals in cells infected with HIV-IN-EGFP virions (*Albanese et al., 2008*) and non-infected control cells. PLA signals in non-infected cells were comparable to those in S9.6-alone and GFP-alone single antibody-negative controls; however, PLA signals significantly increased upon HIV-1 infection (*Figure 5G* and *Figure 5—figure supplement 1E*). Our data suggest that the HIV-1 frequently targets R-loop-rich regions for viral genome integration by physically binding of HIV-1 integrase proteins to R-loop structures on the host genome.

## Discussion

In this study, we found that HIV-1 preferentially integrated into regions rich in R-loops, suggesting that R-loops are novel host factors that contribute to HIV-1 integration site selection. In our bioinformatics analysis, host cellular R-loops were induced by VSV-G-pseudotyped HIV-1 infection and widespread in host genomic regions. Using our R-loop-inducible cell models, R-loop formation, and not necessarily the transcriptional activity itself, was found to be important for HIV-1 integration site determination. In addition, HIV-1 integrase proteins favor physical binding with R-loops in vitro, and interact with host genomic R-loops in HIV-1-infected cells. These results demonstrate that HIV-1 exploits and frequently targets the host genomic R-loop regions for successful integration and infection.

One possible explanation for why HIV-1 integration shows a preference for host genomic R-loop regions is that the R-loop structure may drive dynamics in the genomic environment and the spatial organization of the genome, resulting in increased accessibility of HIV-1 intasome binding to the target host genomic region. The R-loops display enhancer and insulator chromatin states, that can act as distal regulatory elements by recruiting diverse chromatin binding factors (*Sanz et al., 2016*). This not only allows R-loops to drive dynamics in the genome, but also possibly drives R-loop-mediated integration over long-range genomic regions. R-loop regions exhibit increased chromatin accessibility. In the cellular genome, these structures relieve superhelical stresses and are often associated with open chromatin marks and active enhancers (*Sanz et al., 2016*; *Chédin, 2016*), which are also distributed over HIV-1 integration sites (*Schröder et al., 2002*). In the case of transcription-induced R-loop formation, a guanine-quadruplex (G4) structure can be generated in the non-template DNA strand of the R-loop, which is another contributor to genome architecture (*Lee et al., 2020*). A recent study showed that G4 DNA can influence both productive and latent HIV-1 integration, as well as the potential for reactivation of latent proviruses (*Ajoge et al., 2022*).

Another possible explanation for the preference of HIV-1 integration for host genomic R-loops is that R-loops may play a collaborative role with host factors governing the HIV-1 integration site

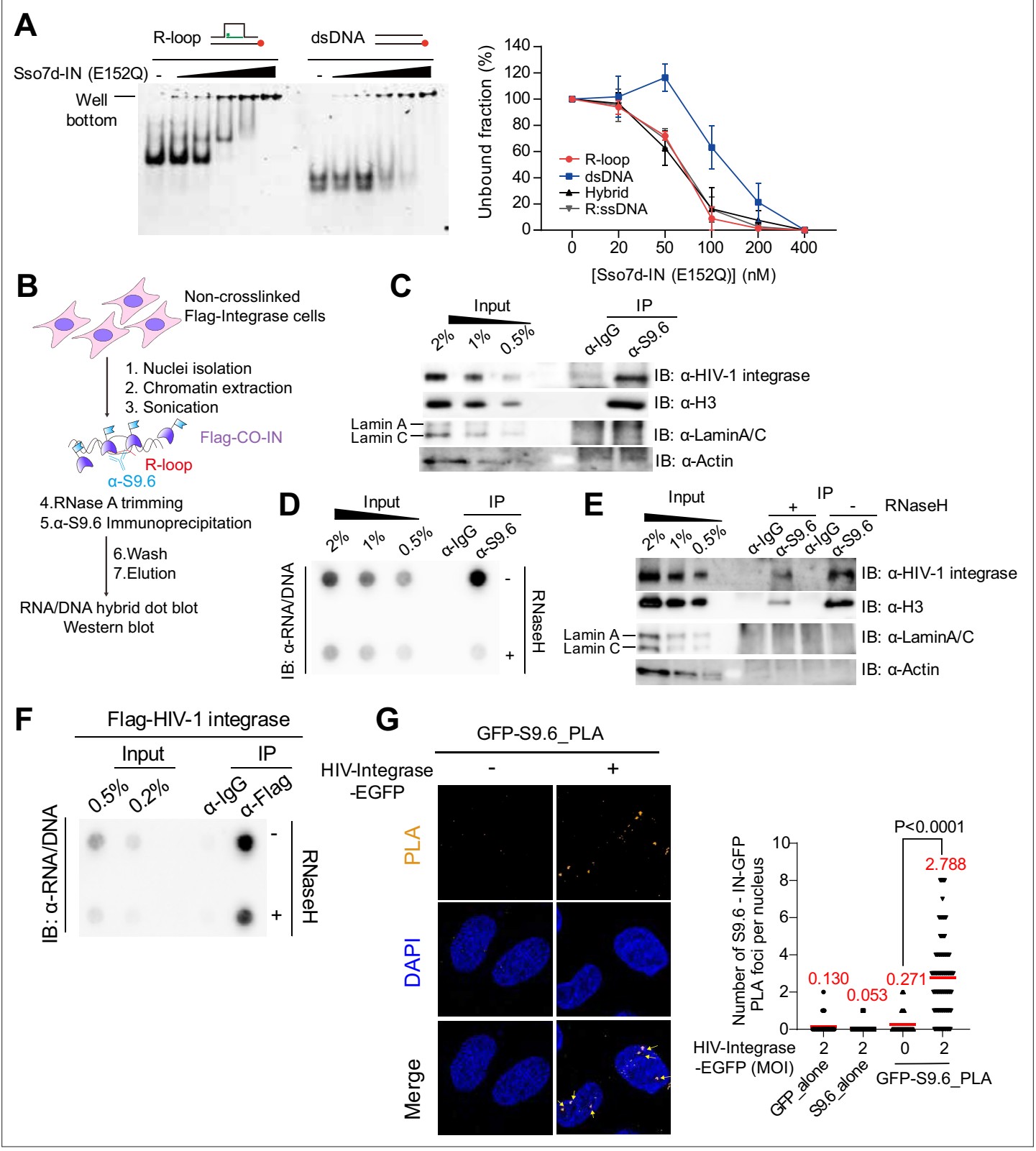

**Figure 5.** HIV-1 integrase proteins directly bind to host genomic R-loops. (**A**) Representative gel images for electrophoretic mobility shift assay (EMSA) of Sso7d-tagged HIV-1-integrase (E152Q) with R-loop and dsDNA, 10 nM nucleic acid substrate was incubated with Sso7d-tagged HIV-1-integrase (E152Q) at 0 nM, 20 nM, 50 nM, 100 nM, 200 nM, and 400 nM (left). Unbound fractions were quantified for EMSA of Sso7d-tagged HIV-1-integrase (E152Q) with different types of substrates (R-loop, dsDNA, R-loop, R:D+ssDNA and Hybrid). Data are presented as the mean ± SEM, n=3 (right).

*Figure 5 continued on next page*

*Figure 5 continued*

(**B**) Summary of the experimental design for R-loop immunoprecipitation using S9.6 antibody in FLAG-tagged HIV-1 integrase protein-expressing HeLa cells. (**C**) Western blotting for HIV-1 integrase protein, H3, and LaminA/C of DNA-RNA hybrid immunoprecipitation using the S9.6 antibody. (**D**) and (**E**) HeLa gDNA input was either untreated (–) or treated (+) with RNase H before enrichment for DNA-RNA hybrids using the S9.6 antibody. gDNA-RNA hybrids were incubated with nuclear extracts depleted of DNA-RNA hybrids with RNase A followed by S9.6 immunoprecipitation. DNA-RNA hybrid dot blot (**D**) and western blot of DNA-RNA hybrid immunoprecipitation, probed with the indicated antibodies (**E**). (**F**) DNA-RNA hybrid dot blot of FLAG antibody-immunoprecipitated nucleic acid extracts. Where indicated, nucleic acid extracts were untreated (–) or treated (+) with RNase H before probing with the S9.6 antibodies. (**G**) Representative images of the proximity-ligation assay (PLA) between GFP and S9.6 antibodies in HIV-IN-EGFP virion-infected HeLa cells at 6 hpi. Cells were subjected to PLA (orange) and co-stained with DAPI (blue). PLA puncta in the nucleus is indicated by the yellow arrows. Quantification analysis of the number of PLA foci per nucleus (left). GFP_alone and S9.6_alone were used as single-antibody controls from HIV-IN-EGFP virion-infected HeLa cells (right). The mean value for each data point is indicated by the red line. p-value was calculated using a two-tailed unpaired *t*-test (n>50).

The online version of this article includes the following source data and figure supplement(s) for figure 5:

**Source data 1.** Original gel image for *Figure 5A*, indicating the relevant bands and treatments.

**Source data 2.** Original files for electrophoretic mobility shift assay (EMSA) analysis are displayed in *Figure 5A*.

**Source data 3.** Original western blots for *Figure 5C*, indicate the relevant bands and treatments.

**Source data 4.** Original files for western blot analysis are displayed in *Figure 5C*.

**Source data 5.** Original western blots for *Figure 5D*, indicate the relevant bands and treatments.

**Source data 6.** Original files for western blot analysis are displayed in *Figure 5D*.

**Source data 7.** Original western blots for *Figure 5E*, indicate the relevant bands and treatments.

**Source data 8.** Original files for western blot analysis are displayed in *Figure 5E*.

**Source data 9.** Original dot blots for *Figure 5F*, indicate the relevant dots and treatments.

**Source data 10.** Original files for dot blot analysis are displayed in *Figure 5F*.

**Figure supplement 1.** HIV-1 integrase proteins directly bind to host genomic R-loops.

**Figure supplement 1—source data 1.** Original dot blots for *Figure 5—figure supplement 1B*, indicating the relevant dots and treatments.

**Figure supplement 1—source data 2.** Original files for dot blot analysis are displayed in *Figure 5—figure supplement 1B*.

**Figure supplement 1—source data 3.** Original western blots for *Figure 5—figure supplement 1D*, indicating the relevant bands and treatments.

**Figure supplement 1—source data 4.** Original files for western blot analysis are displayed in *Figure 5—figure supplement 1D*.

---

selection. Cellular R-loops are recognized and regulated by numerous cellular proteins (*Cristini et al., 2018*; *Mosler et al., 2021*). LEDGF/p75 (*Sowd et al., 2016*; *Cherepanov et al., 2003*; *Schrijvers et al., 2012*) and CPSF6 (*Achuthan et al., 2018*; *Sowd et al., 2016*) are two decisive host factors that direct HIV-1 integration by interacting with integrase or trafficking the viral preintegration complex towards the nuclear interior (*Achuthan et al., 2018*; *Sowd et al., 2016*). In fact, these host factors have recently been identified as potential R-loop-binding proteins in DNA-RNA interactome analysis (*Cristini et al., 2018*) and R-loop proximity proteomics (*Mosler et al., 2021*), respectively. R-loops are tightly regulated by DNA damage response proteins (*Stirling and Hieter, 2017*) and the DNA repair machinery plays an important role in the HIV-1 integration process (*Lesbats et al., 2016*). For example, the Fanconi anemia pathway (*García-Rubio et al., 2015*; *Giannini et al., 2020*), a well-known R-loop regulatory pathway, has been recently proposed as an HIV-1 integration regulatory factor exploited by HIV-1 (*Fu et al., 2022*). Considering theses previous studies and our current findings, we propose R-loops as another putative host factor driving HIV-1 integration site determination, and a possible intermediate regulator of HIV-1 integration site selection by such host proteins.

Our data showed that HIV-1 integrase proteins physically interacted with genomic R-loops in vitro and in cells. Recent advancements in cryogenic electron microscopy (cryo-EM) technology have revealed the conformational characteristics of the target DNA during retroviral integration (*Jóźwik et al., 2022*; *Ballandras-Colas et al., 2022*). During retroviral integration, the target DNA undergoes a transition of its conformation from B-form to A-form. R-loops, which represent intermediates between the B-form DNA and A-form RNA conformations (*Jóźwik et al., 2022*), may have an intrinsic preferential binding ability to retroviral intasomes over other nucleic acid structures.

Viruses often take advantage of various host factors, and targeting the viral components that manipulate the host cellular environment can be an effective strategy for antiviral therapy. Our study showed the host genomic R-loops significantly accumulate shortly after HIV-1 infection. Thus, it is

possible that virion-associated HIV-1 proteins are responsible for inducing these R-loops. For instance, the HIV-1 accessory protein Vpr causes genomic damage (*Li et al., 2020a*) and transcriptomic changes during the early stages post-infection (*Bauby et al., 2021*), both of which can lead to in cis and in trans-R-loop formation (*Petermann et al., 2022*). Another HIV-1 accessory protein, Vif, counteracts the host antiviral factor, APOBEC3 (*Stopak et al., 2003*; *Kmiec and Kirchhoff, 2024*), which was recently found to regulates cellular R-loop levels (*McCann et al., 2021*). Identifying the HIV-1 components responsible for inducing host cellular R-loops, elucidating the mechanism by which they induce genome-wide R-loop formation, and contributing to successful viral integration into selective genomic regions, are areas for further research.

Although most HIV-1 integration occurs in genic regions (*Einkauf et al., 2022*; *Schröder et al., 2002*), HIV-1 proviruses are also found in non-genic regions (*Yukl et al., 2018*), and understanding these 'transcriptionally silent' proviruses is critical for developing strategies to completely eliminate HIV-1. In HIV-1 elite controllers, who suppress viral gene expression to undetectable levels, HIV-1 proviruses in heterochromatic regions are not eliminated but selected by the immune system (*Jiang et al., 2020*). Moreover, proviruses with lower expression levels can persist in the host genome even during antiretroviral therapy (*Einkauf et al., 2022*). However, the mechanism by which HIV-1 targets gene-silent regions for 'invisible' integration remains unclear. Our study has revealed that R-loops are enriched in both genic and non-genic regions during HIV-1 infection, and that the virus preferentially targets these R-loops for integration. We propose that R-loops, particularly those enriched in non-genic regions, may represent the mechanism by which the virus achieves 'invisible' and permanent infection.

## Materials and methods

### Cell culture

HeLa and HEK293T cells were cultured in Dulbecco's modified Eagle's medium (Gibco) supplemented with 10% (v/v) fetal bovine serum (FBS, Cytiva), an antibiotic mixture (100 units/ml penicillin–streptomycin, Gibco), and 1% (v/v) GlutaMAX-I (Gibco). Jurkat cells were cultured in Roswell Park Memorial Institute (RPMI) 1640 medium (ATCC) supplemented with 10% (v/v) FBS (Cytiva). Cells were incubated at 37 °C and 5% $CO_2$.

### Virus production and infection

VSV-G-pseudotyped HIV-1 virus stocks were prepared by performing standard polyethylenimine-mediated transfection of HEK293T monolayers with pNL4-3 ΔEnv EGFP (NIH AIDS Reagent Program 11100) or pNL4-3. Luc.R-E (NIH AIDS Reagent Program, 3418) along with pVSV-G at a ratio of 5:1. HIV-IN-EGFP virions were produced as previously described (*Albanese et al., 2008*) by performing polyethylenimine-mediated transfection of HEK293T cells with 6 µg of pVpr-IN-EGFP, 6 µg of HIV-1 NL4-3 non-infectious molecular clone (pD64E; NIH AIDS Reagent Program 10180), and 1 µg of pVSV-G. The cells were incubated for 4 hr before the medium was replaced with a fresh complete medium. Virion-containing supernatants were collected after 48 hr, filtered through a 0.45 µm syringe filter, and pelleted using the Lenti-X Concentrator (631232; Clontech) according to the manufacturer's instructions. The multiplicity of infection (MOI) of virus stocks was determined by transducing a known number of HeLa cells with a known amount of virus particles and then counting GFP-positive cells using flow cytometry. For luciferase reporter HIV-1 virus, the HIV-1 p24 antigen content in viral stock was quantified using the HIV1 p24 ELISA kit (Abcam, ab218268), according to the manufacturer's instruction. For virus infection, HeLa cells were seeded at a density of $0.5–4\times10^5$ cells/mL on the day before infection. The culture medium was replaced with fresh complete culture medium 2 hpi. The infected cells were washed twice with PBS and harvested at the indicated time points. Jurkat cells were seeded at a density of $1\times10^6$ cells/mL and inoculated with 300 ng/p24 capsid antigen. The plates were centrifuged at 1000 *g* at 30 °C for 1 hr. The medium was replaced with fresh RPMI 2 hr after infection.

### Primary cell isolation, culture, T cell activation, and infection

For CD4[+] T cells isolation, human PBMC (ST70025, STEMCELL Technologies) was mixed and incubated with MACS CD4 MicroBeads (130-045-101, Miltenyi Biotec) and FITC-conjugated mouse

anti-CD4 (561005, BD Bioscience) according to the manufacturer's instructions. Then the CD4[+] T cells were enriched by using LS Columns (130-042-401, Miltenyi Biotec) and MidiMACS Separator (130-042-302, Miltenyi Biotec). The efficiency of magnetic separation was analyzed by using Flow-Activated Cell Sorter Canto II (BD Bioscience) and Flowjo software (Flowjo).

CD4[+] T cells were cultured in Roswell Park Memorial Institute (RPMI) 1640 medium (Gibco), supplemented with 10% (v/v) fetal bovine serum (FBS, Cytiva), an antibiotic mixture (100 units/ml penicillin-streptomycin, Gibco), 1% (v/v) GlutaMAX-I (Gibco), and 20 ng/ml of IL-2 (PHC0026, Gibco), left in resting state or activated with Dynabeads Human T-Activator CD3/CD28 (1161D, Thermo Fisher Scientific) for 72 hr. CD4[+] T cells activation efficiency was assessed by staining cells with FITC-conjugated mouse anti-CD25 (340694, BD Bioscience) and APC-conjugated mouse anti-CD69 (130-114-046, Miltenyi Biotec) and using Flow-Activated Cell Sorter Canto II (BD Bioscience) and Flowjo software (Flowjo).

Purified and activated CD4[+] T cells were seeded at a density of 1×10[6] cells/mL and inoculated with 600 ng/p24 capsid antigen in the presence of polybrene. The plates were centrifuged at 1000 *g* at 30 °C for 1 hr. The medium was replaced with fresh RPMI 2 hr after infection.

## DRIP-qPCR

DRIP was performed as described for the construction of the DRIPc-seq library. After the elution of isolated complexes, nucleic acids were purified using the standard phenol-chloroform extract method and used for qPCR. S6 Table presents details of the primer sequences used for DRIP-qPCR analysis.

## RNA-seq library construction

For RNA-seq, HeLa cells were infected with VSV-G-pseudotyped HIV-1 NL4-3 ΔEnv EGFP virus at an MOI of 0.6 and harvested at 0, 3, 6, and 12 hpi. Sequencing was performed with biological replicates. Total RNA was extracted using TRIzol reagent (Invitrogen), according to the manufacturer's instructions. An mRNA sequencing library was constructed using Illumina adaptors harboring p5 and p7 sequences and Rd1 SP and Rd2 SP sequences. Sequencing was performed using the HiSeq2500 system (Illumina).

## Luciferase assay

HeLa cells infected with VSV-G-pseudotyped pNL4-3.Luc.R-E HIV-1 viruses were harvested at 48 hpi, and luminescence was measured using the Dual-Luciferase Reporter Assay System (Promega) according to the manufacturer's instructions. Briefly, 250 µl of passive lysis buffer was used to lyse cells for each sample, 20 µl of the lysate was mixed with 100 µl of the Luciferase Assay Reagent II, and the luminescence of firefly luciferase was measured using a microplate luminometer (Berthold). The luminescence signals were normalized with total protein content, measured by BCA assay.

## Quantitative real-time PCR (qPCR)

For RT (reverse transcription)-qPCR, 1 µg of RNA was reverse-transcribed using the ReverTra Ace qPCR RT Kit (TOYOBO) following the manufacturer's instructions. For qPCR, DNA extracts were prepared using a DNA purification kit (Qiagen, 51106) according to the manufacturer's instructions. Equivalent amounts of purified gDNA from each sample were analyzed using qPCR. qPCR was performed using TOPreal qPCR PreMIX (Enzynomics, RT500M). The reactions were performed in duplicate or triplicate for technical replicates. PCR was performed using the iCycler iQ real-time PCR detection system (Bio-Rad). All the primers used for qPCR are listed in the S6 Table.

## DRIPc-seq library construction

DRIP followed by library preparation, next-generation sequencing, and peak calling were performed as described earlier (*Sanz and Chédin, 2019*). Briefly, the corresponding cells were harvested and their gDNA was extracted. The extracted nucleic acids were fragmented using a restriction enzyme cocktail with BsrB I (NEB, R0102S), HindIII (NEB, R0136L), Xba I (NEB, R0145L), and EcoRI (NEB, R3101L) overnight at 37 °C. Half of the fragmented nucleic acids were digested with RNase H (New England Biolabs) overnight at 37 °C to serve as a negative control. The digested nucleic acids were cleaned using standard phenol-chloroform extraction and resuspended in DNase/RNase-free water. DNA-RNA hybrids were immunoprecipitated from total nucleic acids using mouse anti-DNA-RNA

hybrid S9.6 (Kerafast, ENH001) DRIP binding buffer and incubated overnight at 4 °C. Dynabeads Protein A (Invitrogen, 10001D) was used to pull down the DNA-antibody complexes by incubation for 4 hr at 4 °C. The isolated complexes were washed twice with DRIP binding buffer before elution. For elution, the isolated complexes were incubated in an elution buffer containing proteinase K for 45 min at 55 °C. Subsequently, DNA was purified using the standard phenol-chloroform extract method and subjected to DNase I (Takara, 2270 B) treatment and reverse transcription for DRIPc-seq library construction. DRIPc-seq was performed in biological replicates. S5 Table shows details of the oligo-nucleotides used for DRIPc-seq library construction. DRIPc-seq libraries were analyzed using 150 bp paired-end sequencing on a HiSeqX Illumina instrument.

## Immunofluorescence microscopy

For immunofluorescence assays of S9.6 nuclear signals, when indicated, the cells were pre-extracted with cold 0.5% NP-40 for 3 min on ice. Cells were fixed with 100% ice-cold methanol for 10 min on ice and then incubated with 100% ice-cold acetone for 1 min. The slides were washed three times with 1x PBS and incubated with or without 60 U/mL RNase H (M0297S, NEB) at 37 °C for 36 hr or left untreated. The slides were subsequently briefly rinsed thrice with 2% BSA/0.05% Tween (in PBS) and incubated with mouse anti-DNA-RNA hybrid S9.6 (Kerafast, ENH001; 1:100) and rabbit anti-nucleolin (Abcam, ab22758; 1:300) in 2% BSA/0.05% Tween (in PBS) for 4 hr at 4 °C. The slides were then washed three times with 2% BSA/0.05% Tween (in PBS) and incubated with goat anti-rabbit AlexaFluor-488-conjugated (Invitrogen, A-11008) and goat anti-mouse AlexaFluor-568-conjugated (Molecular Probes, A11004) secondary antibodies (1:200) for 2 hr at room temperature. The slides were then washed three times with 2% BSA/0.05% Tween (in PBS) and mounted using the ProLong Gold AntiFade reagent (Invitrogen). Images were obtained using an inverted microscope Nikon Eclipse Ti2, equipped with a 1.45 numerical aperture, plan apochromat lambda 100×oil objective, and a scientific complementary metal–oxide–semiconductor camera (Photometrics prime 95 B 25 mm). For each field of view, images were obtained with DAPI395, GFP488, and Alexa594 channels using the NIS-Elements software. For quantification analysis, binary masks of nuclei and nucleoli were generated using the ROI manager and auto-local thresholding using the ImageJ software. The intensity of nuclear signals for DNA-RNA hybrids and nucleolin was then quantified. The final DNA-RNA hybrid signals in the nucleus were calculated by subtracting the nucleolin signals from the DNA-RNA hybrid signals.

## pgR-rich and -poor cell line generation with piggyBac transposition

We adapted and modified an elegantly designed episomal system that induces defined R-loops with controlled transcription levels (*Hamperl et al., 2017*) for R-loop-forming or non-R-loop-forming sequence subcloning into the piggyBac transposon vector. HeLa cells were seeded at a density of $5×10^4$ cells/ml in a six-well plate. The next day, cells were transfected with 0.2 μg of Super PiggyBac Transposase Expression Vector (System Biosciences, PB210PA-1) and 0.2, 1, or 2 μg of transposon vectors with appropriate 'cargo' sub cloned using Lipofectamine 3000 (Invitrogen) according to the manufacturer's instructions. After 3 d, the cells were treated with 10 μg/ml blasticidin S (Gibco, A1113903) for selection. Cells with positive integrants for more than 7 d were validated using immu-noblotting or RT-qPCR following treatment with DOX. Jurkat cells were seeded at a density of $8×10^5$ cells/ml in a six-well plate and transfected with 0.2 μg of transposase and 1 μg of corresponding transposon vectors with Lipofectamine 3000, like HeLa cells. After 3 d, the cells were treated with 10 μg/ml blasticidin S (Gibco, A1113903) for selection. For each passage, cells were cushioned onto Ficoll-Pacque (Cytiva, 17144002) to separate live cells from dead cell debris. The cells over the cushion were washed with PBS and incubated in a cell culture medium with 10 μg/ml of blasticidin for further selection for at least 14 d. Cells with positive integrants were validated by immunoblotting after treat-ment with DOX. Quantification of successfully integrated piggyBac transposons was performed using a piggyBac qPCR copy number kit (System Biosciences, PBC100A-1).

## HIV-1 integration site sequencing library construction

HIV-1 integration site sequencing library construction was performed as described earlier (*Achuthan et al., 2018*; *Sowd et al., 2016*). Summarily, HeLa cells and primary CD4+ T cells were infected with VSV-G-pseudotyped HIV-1 NL4-3 ΔEnv EGFP virus at an MOI of 0.6 and harvested 5 d post-infection. gDNA was isolated using a DNA purification kit (Qiagen, 51106), according to the manufacturer's

instructions. gDNA (10 µg) was digested overnight at 37 °C with 100 U each of the restriction endo-nucleases MseI (NEB, R0525L) and BglII (NEB, R0144L). Linker oligonucleotides, which were compatible for ligation with the MseI-generated DNA ends, were ligated with gDNA overnight at 12 °C in reactions containing 1.5 µM ligated linker, 1 µg fragmented DNA, and 800 U T4 DNA ligase (NEB, M0202S). Viral LTR–host DNA junctions were amplified using semi-nested PCR with a unique linker-specific primer and LTR primers. The second round of PCR was carried out with primers binding to the LTR and the linkers for next-generation sequencing. Two PCRs were performed in parallel for the first round of PCR and five PCRs were performed in parallel for the second round of PCR to enhance library diversity. S7 Table presents details of the oligonucleotides used for HIV-1 integration site sequencing library construction. HIV-1 integration site sequencing was performed in biological replicates. Integration site libraries were analyzed using 150 bp paired-end sequencing on a HiSeqX Illumina instrument.

## Co-immunoprecipitation of DNA-RNA hybrid

DNA-RNA hybrid immunoprecipitation was performed as described earlier (*Cristini et al., 2018*). Summarily, non-crosslinked HeLa cells transfected with the pFlag-IN codon-optimized plasmid were lysed in 85 mM KCl, 5 mM PIPES (pH 8.0), and 0.5% NP-40 for 10 min on ice, and then, the lysates were centrifuged at 750 *g* for 5 min to pellet the nuclei. The pelleted nuclei were resuspended in sodium deoxycholate, SDS, and sodium lauroyl sarcosinate in RSB buffer and were sonicated for 10 min (Diagenode Bioruptor). Extracts were then diluted (1:4 in RSB +T buffer) and subjected to immunoprecipitation with the S9.6 antibody overnight at 4 °C. Antibody-bound complexes were incubated with Protein A Dynabeads (Invitrogen) for 4 hr at 4 °C for immunoprecipitation. Normal mouse IgG antibodies (Santa Cruz, sc-2025) were used as negative controls. RNase A (Thermo Scientific, EN0531) was added during immunoprecipitation at 0.1 ng RNase A per µg gDNA. Beads were washed four times with RSB +T; twice with RSB, and eluted either in 2x LDS (Novex, NP0007), 100 mM DTT for 10 min at 70 °C (for western blot), or 1% SDS and 0.1 M NaHCO$_3$ for 30 min at room temperature (for DNA-RNA hybrid dot blot).

For co-immunoprecipitation of DNA-RNA hybrids with RNase H treatment, gDNA containing RNA-DNA hybrids were isolated from HeLa cells transfected with a pFlag-IN codon-optimized plasmid using a QIAmp DNA Mini Kit (Qiagen, 51304). gDNA was sonicated for 10 min (Diagenode Bioruptor) and then treated with 5.5 U RNase H (NEB, M0297) per µg of DNA overnight at 37 °C. A fraction of gDNA was stored as 'nucleic acid input' for dot blot analysis. gDNA was cleaned using standard phenol-chloroform extraction, resuspended in DNase/RNase-free water, enriched for DNA-RNA hybrids using immunoprecipitation with the S9.6 antibody (overnight at 4 °C), isolated with Protein A Dynabeads (Invitrogen; 4 hr at 4 °C), washed thrice with RSB +T. The immunoprecipitated complexes were incubated with nuclear extracts of HeLa cells transfected with the pFlag-IN codon-optimized plasmid for 2 hr at 4 °C with diluted HeLa nuclear extracts. The cell lysate-containing proteins were pre-treated with 0.1 mg/ml RNase A (Thermo Scientific, EN0531) for 1 hr at 37°C to degrade all RNA-DNA hybrids, and the excess of RNase A was blocked by adding 200 U of SUPERase in RNase inhibitor (Invitrogen, AM2694) for immunoprecipitation. In addition, 100 µL fraction of diluted and RNase A pre-treated extracts prior to immunoprecipitation was stored as 'protein input' for western blotting. Beads were washed four times with RSB +T; twice with RSB, and eluted either in 2x LDS (Novex, NP0007), 100 mM DTT for 10 min at 70 °C (for western blot), or 1% SDS, and 0.1 M NaHCO$_3$ for 30 min at room temperature (for DNA-RNA hybrid dot blot).

## Recombinant Sso7d-IN protein purification

Sso7d-integrase active site mutant E152Q was expressed in *Escherichia coli* BL21-AI and purified essentially as previously described (*Passos et al., 2017*). Briefly, Sso7d-IN (E152Q) expressed BL21-AI cells were lysis in lysis buffer (20 mM HEPES pH 7.5, 2 mM 2-mercaptoethanol, 1 M NaCl, 10% (w/v) glycerol, 20 mM imidazole, 1 mg RNase A, and 1000 U DNase I) and purified by nickel affinity chromatography (Qiagen, 30210). Proteins were first loaded on a HeparinHP column (GE Healthcare) equilibrated with equilibrated with 20 mM Tris, pH 8.0, 0.5 mM TCEP, 200 mM NaCl, and 10% glycerol for anion exchange chromatography prior to the size exclusion chromatography. Proteins were eluted with a linear gradient of NaCl from 200 mM to 1 M. Eluted fractions were pooled and then separated on a Superdex-200 PC 10/300 GL column (GE Healthcare) equilibrated with 20 mM Tris pH 8.0, 0.5 mM TCEP, 500 mM NaCl and 6% (w/v) glycerol. The purified protein was concentrated to

0.6 mg/ml using an Amicon centrifugal contentators (EMD Millipore), flash-frozen in liquid nitrogen, and stored at –80 °C.

## Electrophoretic mobility shift assay for R-loop binding of Sso7d-IN

To test the binding affinity of Sso7d-tagged HIV-1-IN to different types of nucleic acid substrates, we prepared R-loop, dsDNA, RNA-DNA hybrid with exposed ssDNA (R:D+ssDNA), and RNA-DNA hybrid (Hybrid) by annealing different combinations of Cy3, Cy5 or non-labeled oligonucleotides following the previous protocol (*Nguyen et al., 2017*; *Kang et al., 2021*). 10 nM of DNA substrate was incubated with Sso7d-IN at different concentrations in assembly buffer (20 mM HEPES pH 7.5, 5 mM $CaCl_2$, 8 mM 2-mercaptoethanol, 4 uM $ZnCl_2$, 100 mM NaCl, 25% (w/v) glycerol and 50 mM 3-(Benzyldimethylammonio) propanesulfonate (NDSB-256)), for 1 hr at 30 °C then incubated for 15 min on ice. All the reactants were run on 4.5% non-denaturing PAGE in 1x TBE and then Cy3 or Cy5 fluorescence signal was imaged by ChemiDoc MP imaging system (Bio-Rad). S8 Table presents details of the oligonucleotide sequence used for EMSA.

## PLA

For PLA, HeLa cells were grown on coverslips and infected with HIV-IN-EGFP virions. At 6 hpi, cells were pre-extracted with cold 0.5% NP-40 for 3 min on ice. The cells were fixed with 4% paraformaldehyde in PBS for 15 min at 4 °C. The cells were then blocked with 1×blocking solution (Merck, DUO92102) for 1 hr at 37 °C in a humidity chamber. After blocking, cells were incubated with the following primary antibodies overnight at 4 °C for S9.6-HIV-1-IN_PLA: mouse anti-DNA-RNA hybrid S9.6 (1:250; Kerafast, ENH001) and rabbit anti-GFP (1:500; Abcam, ab6556). The following day, after washing with once with buffer A twice (Merck, DUO92102), cells were incubated with pre-mixed Duolink PLA plus (anti-mouse) and PLA minus probes (anti-rabbit) antibodies for 1 hr at 37 °C. The subsequent steps in the proximal ligation assay were performed using the Duolink PLA Fluorescence kit (Sigma) according to the manufacturer's instructions. To obtain images, the mounted specimens were visually scanned and representative images were acquired using a Zeiss LSM 710 laser scanning confocal microscope (Carl Zeiss). The number of intranuclear PLA puncta was quantified using the ImageJ software. For each biological replicate and experiment, a PLA with a single antibody was performed as a negative control under the same conditions.

## DRIPc-seq data processing and peak calling

DRIPc-seq reads were quality-controlled using FastQC v0.11.9 (*Andrews, 2010*), and sequencing adapters were trimmed using Trim Galore! v0.6.6 (*Krueger et al., 2021*) based on Cutadapt v2.8 (*Martin, 2011*). Trimmed reads were aligned to the hg38 reference genome using bwa v0.7.17-r1188 (*Li and Durbin, 2009*). Read deduplication and peak calling were performed using MACS v2.2.7.1 (*Zhang et al., 2008*). Because R-loops appear as both narrow and broad peaks in DRIPc-seq read alignment owing to its variable length, two independent 'MACS2 call peak' runs were performed for narrow and broad peak calling. The narrow and broad peaks were merged using Bedtools v2.26.0 (*Quinlan and Hall, 2010*). To increase the sensitivity of DRIPc-seq peak identification, peaks were called after pooling the two biological replicates of the DRIPc-seq sequencing data for each condition.

## Consensus R-loop peak calling

The R-loop peaks at 0, 3, 6, and 12 hpi were first merged using 'bedtools merge' to create a universal set of R-loop peaks across time points (n=46542). Then, each of the universal R-loop peaks was tested for overlap with the R-loop peaks for 0, 3, 6, and 12 hpi using 'bedtools intersect.' In all, 9,190, 21,403, 33,544, and 9,941 peaks overlapped with 0, 3, 6, and 12 hpi R-loop peaks, respectively. For CD4 cells, we identified a universal R-loop set consisting of 3,928 R-loops, and among them, 737, 722, 1,796, and 2,766 peaks overlapped with 0, 3, 6, and 12hpi R-loop peaks.

## HIV-1 integration site sequencing data processing

Quality control of HIV-1 integration site-sequencing reads was performed using FastQC v0.11.9. To discard primers and linkers specific for integration site-sequencing from reads, we used Cutadapt v2.8 with the following option: '-u 49 U 38 `--minimum-length 36 --pair-filter any --action trim` -q0,0 –a linker -A TGCTAGAGATTTTCCACACTGACTGGGTCTGAGGG -A GGGTCTGAGGG

`--no-indels --overlap 12`.' This allowed the first position of the read alignment to directly represent the genomic position of HIV-1 integration. Processed reads were aligned to the hg38 reference genome using bwa v0.7.17-r1188, and integration sites were identified using an in-house Python script. Genomic positions supported by more than five read alignments were regarded as HIV-1 integration sites. For Jurkat cells, we adopted integration site sequencing data of HIV-1 infected wild-type Jurkat cells from SRR12322252 (*Li et al., 2020b*).

## Co-localization analysis of R-loops and integration sites

Enrichment of integration sites near the R-loop peaks was tested using a randomized permutation test. Randomized R-loop peaks were generated using 'bedtools shuffle' command, thus preserving the number and the length distribution of the R-loop peaks during the randomization process. Similarly, integration sites were randomized using the 'bedtools shuffle' command. Randomization was performed 100 times. ENCODE blacklist regions (*Amemiya et al., 2019*) were excluded while shuffling the R-loops and integration sites to exclude inaccessible genomic regions from the analysis. For each of the observed (or randomized) integration sites, the closest observed (or randomized) R-loop peak and the corresponding genomic distance were identified using the 'bedtools closest' command. The distribution of the genomic distances was displayed to show the local enrichment of integration sites in terms of the increased proportion of integration sites within the 30 kb window centered on R-loops compared to their randomized counterparts.

## DNA plasmid construction and transfection

R-loop-forming mAIRN and non-R-loop forming ECPF sequences were subcloned from pSH26 and pSH36 plasmids, which were generously provided by Prof. Karlene A. Cimprich, into the piggyBac transposon vector, where the tet operator sequences were located upstream of the minimal CMV promoter. The pFlag-IN codon-optimized plasmid and pVpr-IN-EGFP were kindly provided by Prof. A. Engelman and Prof. Anna Cereseto, respectively. Lipofectamine 3000 (Invitrogen) transfection reagent was used for the transfection of all plasmids into cells, according to the manufacturer's protocol.

## DNA-RNA hybrid dot blotting

Total gDNA was extracted using the QIAmp DNA Mini Kit (Qiagen, 51304) according to the manufacturer's instructions. gDNA (1.2 µg) was treated with 2 U RNase H (NEB, M2097) per µg of gDNA for 4 h at 37 °C, with half of the sample left untreated but denatured. Half of the DNA sample was probed with S9.6 an antibody (1:1000), and the other half was probed with an anti-ssDNA antibody (MAB3034, Millipore, 1:10000).

## Immunoblotting

Cells were lysed using RIPA buffer (50 mM Tris, 150 mM sodium chloride, 0.5% sodium deoxycholate, 0.1% SDS, and 1.0% NP-40) supplemented with 10 µM leupeptin (Sigma-Aldrich) and 1 mM phenyl-methanesulfonyl fluoride (Sigma-Aldrich) and boiled at 98 °C for 10 min with SDS sample buffer prior to SDS-PAGE. The primary antibodies used were mouse monoclonal anti-FLAG M2 (Sigma, F3165), monoclonal mouse anti-HSC70 (Abcam, ab2788), polyclonal rabbit anti-histone H3 (tri methyl K4) antibody (Abcam, ab8580), monoclonal mouse anti- HIV-1 Integrase (Santa Cruz, sc-69721), rabbit anti-LaminA/C antibody (Cell Signaling, 2032), and monoclonal mouse anti-Actin (Invitrogen, MA1-744). All primary antibodies were used at a dilution of 1:1000 for western blotting. Peroxidase-conjugated anti-mouse IgG (115-035-062) and anti-rabbit IgG (111-035-003; both Jackson Laboratories) were used as secondary antibodies at 1:5000 dilution. Signals were detected using the SuperSignal West Pico chemiluminescence kit (Thermo Fisher Scientific).

## RNA-seq data processing

RNA-seq reads were quality-controlled and adapter-trimmed as in DRIPc-seq processing. To quantify the expression levels of protein-coding genes, processed reads were aligned to the hg38 reference genome with GENCODE v37 gene annotation (*Frankish et al., 2021*) using STAR v2.7.3a (*Dobin et al., 2013*). Gene expression quantification was performed using RSEM v1.3.1. To quantify the expression levels of transposable elements (TEs), we used TEtranscripts v2.2.1 (*Jin et al., 2015*). Processed reads were first aligned to the hg38 reference genome using GENCODE v37 and RepeatMasker

TE annotation using STAR v2.7.3a. In this case, STAR options were modified as follows to utilize multimapping reads in downstream analyses: '`--outFilterMultimapNmax 100 --winAnchor-MultimapNmax 100 --outMultimapperOrder random --runRNGseed 77 --outSAMmult-Nmax 1 --outFilterType BySJout --alignSJoverhangMin 8 --alignSJDBoverhangMin 1 --alignIntronMin 20 --alignIntronMax 1000000 --alignMatesGapMax 1000000.`' Expression levels of TEs were quantified as read counts with the 'TEcount' command.

## Genome annotations

All bioinformatic analyses were performed using the hg38 reference genome and GENCODE v37 gene annotation. Promoters were defined as a 2 kb region centered at the transcription start sites of the APPRIS principal isoform of protein-coding genes. TTS regions were defined as the 2 kb region centered at the 3′ terminals of protein-coding transcripts. CpG island annotations were downloaded from the UCSC table browser. CpG shores were defined as 2 kb regions flanking CpG islands, excluding the regions overlapping with CpG islands. Similarly, CpG shelves were defined as 2 kb regions flanking the stretch of CpG islands and shores while excluding the regions overlapping with CpG islands and shores. Annotations for LINE, SINE, and LTR were extracted from the RepeatMasker track in the UCSC table browser.

## Identification of viral sequencing reads in DRIPc-seq

To identify sequencing reads originating from the viral genome, we aligned DRIPc-seq reads to a composite reference genome consisting of the human and HIV1 genome (Genbank accession number: AF324493.2) and computed the proportion of the reads mapped to the HIV1 genome.

# Acknowledgements

We are grateful to Prof. Karlene A Cimprich (Standford University) for providing the pSH26 and pSH36 plasmids, Prof. A Engelman (Harvard Medical School) for providing the pFlag-IN codon optimized plasmid, and Prof. Anna Cereseto (University of Trento) for providing pVpr-IN-EGFP. The NL4-3 ΔEnv EGFP and pNL4-3.Luc.R-E- viral plasmids were obtained through the NIH HIV Reagent Program, Division of AIDS, NIAID, NIH. We thank Dr. Sungchul Kim (IBS center for RNA Research) and Seongjin An (Korea University) for their technical support.

# Additional information

### Funding

| Funder | Grant reference number | Author |
| --- | --- | --- |
| Institute for Basic Science | IBS-R008-D1 | Kwangseog Ahn |
| National Research Foundation of Korea | NRF-2020R1A2C3011298 | Kwangseog Ahn |
| National Research Foundation of Korea | NRF-2020R1A5A1018081 | Kwangseog Ahn |

The funders had no role in study design, data collection and interpretation, or the decision to submit the work for publication.

### Author contributions

Kiwon Park, Conceptualization, Data curation, Validation, Investigation, Visualization, Methodology, Writing - original draft, Writing - review and editing; Dohoon Lee, Data curation, Software, Formal analysis, Methodology, Writing - review and editing; Jiseok Jeong, Sungwon Lee, Data curation, Writing - review and editing; Sun Kim, Formal analysis, Supervision, Writing - review and editing; Kwangseog Ahn, Conceptualization, Supervision, Funding acquisition, Writing - review and editing

### Author ORCIDs

Kiwon Park ![ORCID] http://orcid.org/0009-0001-0190-5077
Kwangseog Ahn ![ORCID] https://orcid.org/0000-0002-1015-245X

Reviewer #1 (Public review): https://doi.org/10.7554/eLife.97348.3.sa1
Reviewer #3 (Public review): https://doi.org/10.7554/eLife.97348.3.sa2
Author response https://doi.org/10.7554/eLife.97348.3.sa3

# Additional files

## Supplementary files
• MDAR checklist

## Data availability
All commercial and open source code used to analyse the data in this study are included in the Key Resources Table. Custom code, bioinformatics pipelines and scripts used in this study are accessible from https://github.com/dohlee/hiv1-rloop (copy archived at *Lee, 2024*).

The following datasets were generated:

| Author(s) | Year | Dataset title | Dataset URL | Database and Identifier |
|---|---|---|---|---|
| Park K, Lee D, Kim S, Ahn K | 2024 | Human immunodeficiency virus-1 targets genomic R-loops of the host for integration [DRIPc] | https://www.ncbi.nlm.nih.gov/geo/query/acc.cgi?acc=GSE281025 | NCBI Gene Expression Omnibus, GSE281025 |
| Park K, Lee D, Kim S, Ahn K | 2024 | Human immunodeficiency virus-1 targets genomic R-loops of the host for integration [IS-seq] | https://www.ncbi.nlm.nih.gov/geo/query/acc.cgi?acc=GSE281026 | NCBI Gene Expression Omnibus, GSE281026 |
| Park K, Lee D, Kim S, Ahn K | 2024 | Human immunodeficiency virus-1 targets genomic R-loops of the host for integration [RNA-seq] | https://www.ncbi.nlm.nih.gov/geo/query/acc.cgi?acc=GSE281027 | NCBI Gene Expression Omnibus, GSE281027 |

The following previously published datasets were used:

| Author(s) | Year | Dataset title | Dataset URL | Database and Identifier |
|---|---|---|---|---|
| Li W, Singh PK, Sowd GA, Bedwell GJ, Jang S, Achuthan AV, Wong D, Fadel HJ, Lee KE, KewalRamani VN, Poeschla EM, Herschhorn A, Engelman AN | 2020 | Jurkat_WT_HIV1_S1 | https://www.ncbi.nlm.nih.gov/sra/SRR12322252 | NCBI Sequence Read Archive, SRR12322252 |
| Chen Y, Zhang Y, Wang Y, Zhang L, Brinkman EK, Adam SA, Goldman R, Steensel BV, Ma J, Belmont AS | 2018 | GSM2157131: SON TSA-Seq Condition 1 (TSA only); *Homo sapiens*; OTHER | https://www.ncbi.nlm.nih.gov/sra/?term=SRR3538917 | NCBI Sequence Read Archive, SRR3538917 |
| Chen Y, Zhang Y, Wang Y, Zhang L, Brinkman EK, Adam SA, Goldman R, Steense BV, Ma J, Belmont AS | 2018 | GSM2157132: SON TSA-Seq Condition 1 (TSA only) Second pull down; *Homo sapiens*; OTHER | https://www.ncbi.nlm.nih.gov/sra/?term=SRR3538918 | NCBI Sequence Read Archive, SRR3538918 |

*Continued*

| Author(s) | Year | Dataset title | Dataset URL | Database and Identifier |
|---|---|---|---|---|
| Chen Y, Zhang Y, Wang Y, Zhang L, Brinkman EK, Adam SA, Goldman R, Steensel BV, Ma J, Belmont AS | 2018 | GSM2157133: input_for_ SON TSA-Seq Condition 1 (TSA only) and SON TSA-Seq Condition 1 (TSA only) Second pull down; *Homo sapiens*; OTHER | https://www.ncbi.nlm. nih.gov/sra/?term= SRR3538919 | NCBI Sequence Read Archive, SRR3538919 |
| Chen Y, Zhang Y, Wang Y, Zhang L, Brinkman EK, Adam SA, Goldman R, Steensel BV, Ma J, Belmont AS | 2018 | GSM2157134: SON TSA-Seq Condition 2 (TSA+Sucrose); *Homo sapiens*; OTHER | https://www.ncbi.nlm. nih.gov/sra/?term= SRR3538920 | NCBI Sequence Read Archive, SRR3538920 |
| Meuleman W, Peric-Hupkes D, Kind J, Beaudry JB, Pagie L, Kellis M, Reinders M, Wessels L, Steensel BV | 2013 | Evolutionary conservation of nuclear lamina-genome interactions | https://www.ncbi.nlm. nih.gov/bioproject/? term=GSE22428 | NCBI BioProject, GSE22428 |

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

# Appendix 1

## Appendix 1—key resources table

| Reagent type (species) or resource | Designation | Source or reference | Identifiers | Additional information |
|---|---|---|---|---|
| Sequence-based reagent | P1 Fwd | This paper | | TTATAAGTCAGCCTCCAGGATCAA |
| Sequence-based reagent | P1 Rev | This paper | | TTCAGGTCTAGGCAGTCTGA |
| Sequence-based reagent | P2 Fwd | This paper | | GGACAGATGACAGGGTCGC |
| Sequence-based reagent | P2 Rev | This paper | | ATGAGGAAGACCCCCTCGG |
| Sequence-based reagent | P3 Fwd | This paper | | CTCTGTGTAACGCTGGTGCT |
| Sequence-based reagent | P3 Rev | This paper | | ACACGCTTCTGACCACTAAGG |
| Sequence-based reagent | N1 Fwd | This paper | | TTGGCCCTACTGAATGATTGGT |
| Sequence-based reagent | N1 Rev | This paper | | TTAAGGCATGCTCAGGCGA |
| Sequence-based reagent | N2 Fwd | This paper | | TGAGATTTCAGGTTCCATGATTTG |
| Sequence-based reagent | N2 Rev | This paper | | TGCTCAGTGTTCTAATTTCCCTGT |
| Sequence-based reagent | β-actin Fwd | This paper | | AGAGCTACGAGCTGCCTGAC |
| Sequence-based reagent | β-actin Rev | This paper | | AGCACTGTGTTGGCGTACAG |
| Sequence-based reagent | SH49 (ECFP Fwd) | *Hamperl et al., 2017* | | TGGTTTGTCCAAACTCATCAA |
| Sequence-based reagent | SH40 (mAIRN Fwd) | *Hamperl et al., 2017*; | | CGAGAGAGGCTAAGGGTGAA |
| Sequence-based reagent | SH21 (ECFP/mAIRN Rev) | *Hamperl et al., 2017*; *Braun, 2018* | | ACATGGTCCTGCTGGAGTTC |
| Sequence-based reagent | RT-qPCR P1 (TOR1AIP2) Fwd | This paper | | CCTTGGTCTTTCCCACTTGAGTG |
| Sequence-based reagent | RT-qPCR P1 (TOR1AIP2) Rev | This paper | | GCAGGGTTAAAACCAGCTACTCG |
| Sequence-based reagent | RT-qPCR P2 (DVL1) Fwd | This paper | | GCATAACCGACTCCACCATGTC |
| Sequence-based reagent | RT-qPCR P2 (DVL1) Rev | This paper | | GATGGAGCCAATGTAGATGCCG |
| Sequence-based reagent | RT-qPCR P3 (PKN2) Fwd | This paper | | GCATCACCAACACTAAGTCCACG |
| Sequence-based reagent | RT-qPCR P3 (PKN2) Rev | This paper | | GCTTTTGACCGTCCAGGGACAT |
| Sequence-based reagent | RT-qPCR N2 (CDK5RAP1) Fwd | This paper | | AGAGTGGAAGCAGCCGTGTGTT |
| Sequence-based reagent | RT-qPCR N2 (CDK5RAP1) Rev | This paper | | GATCTTCCTCCGTCTCACCACA |
| Sequence-based reagent | PCR primer 1.0 P5 | *Sanz and Chédin, 2019* | | AATGATACGGCGACCACCGAGATCTAC ACTCTTTCCCTACACGA |
| Sequence-based reagent | PCR primer 2.0 P7 | *Sanz and Chédin, 2019* | | CAAGCAGAAGACGGCATACGAGAT |
| Sequence-based reagent | Index Adapter 1 | Illumina | HeLa 0hpi Input replicate 1; Jurkat 0hpi Input replicate 1 | GATCGGAAGAGCACACGTCTGAACTCC AGTCACATCACGATCTCGTATGCCGTCTTCTGCTTG |

*Appendix 1 Continued on next page*

*Appendix 1 Continued*

| Reagent type (species) or resource | Designation | Source or reference | Identifiers | Additional information |
|---|---|---|---|---|
| Sequence-based reagent | Index Adapter 2 | Illumina | HeLa 3hpi Input replicate 1; Jurkat 3hpi Input replicate 1 | GATCGGAAGAGCACACGTCTGAACTCCAGT CACCGATGTATCTCGTATGCCGTCTTCTGCTTG |
| Sequence-based reagent | Index Adapter 3 | Illumina | HeLa 6hpi Input replicate 1; Jurkat 6hpi Input replicate 1 | GATCGGAAGAGCACACGTCTGAACTCCAGT CACTTAGGCATCTCGTATGCCGTCTTCTGCTTG |
| Sequence-based reagent | Index Adapter 4 | Illumina | HeLa 12hpi Input replicate 1; Jurkat 12hpi Input replicate 1 | GATCGGAAGAGCACACGTCTGAACTCCAGT CACTGACCAATCTCGTATGCCGTCTTCTGCTTG |
| Sequence-based reagent | Index Adapter 5 | Illumina | HeLa 0hpi RNH-IP replicate 1; Jurkat 0hpi RNH-IP replicate 1 | GATCGGAAGAGCACACGTCTGAACTCCAGT CACACAGTGATCTCGTATGCCGTCTTCTGCTTG |
| Sequence-based reagent | Index Adapter 6 | Illumina | HeLa 3hpi RNH-IP replicate 1; Jurkat 3hpi RNH-IP replicate 1 | GATCGGAAGAGCACACGTCTGAACTCCAGTC ACGCCAATATCTCGTATGCCGTCTTCTGCTTG |
| Sequence-based reagent | Index Adapter 7 | Illumina | HeLa 6hpi RNH-IP replicate 1; Jurkat 6hpi RNH-IP replicate 1 | GATCGGAAGAGCACACGTCTGAACTCCAGTC ACCAGATCATCTCGTATGCCGTCTTCTGCTTG |
| Sequence-based reagent | Index Adapter 8 | Illumina | HeLa 12hpi RNH-IP replicate 1; Jurkat 12hpi RNH-IP replicate 1 | GATCGGAAGAGCACACGTCTGAACTCCAGTCAC ACTTGAATCTCGTATGCCGTCTTCTGCTTG |
| Sequence-based reagent | Index Adapter 9 | Illumina | HeLa 0hpi RNH +IP replicate 1; Jurkat 0hpi RNH +IP replicate 1 | GATCGGAAGAGCACACGTCTGAACTCCAGTC ACGATCAGATCTCGTATGCCGTCTTCTGCTTG |
| Sequence-based reagent | Index Adapter 10 | Illumina | HeLa 3hpi RNH +IP replicate 1; Jurkat 3hpi RNH +IP replicate 1 | GATCGGAAGAGCACACGTCTGAACTCCAGT CACTAGCTTATCTCGTATGCCGTCTTCTGCTTG |
| Sequence-based reagent | Index Adapter 11 | Illumina | HeLa 6hpi RNH +IP replicate 1; Jurkat 6hpi RNH +IP replicate 1 | GATCGGAAGAGCACACGTCTGAACTCCAGT CACGGCTACATCTCGTATGCCGTCTTCTGCTTG |
| Sequence-based reagent | Index Adapter 12 | Illumina | HeLa 12hpi RNH +IP replicate 1; Jurkat 12hpi RNH +IP replicate 1 | GATCGGAAGAGCACACGTCTGAACTCCAG TCACCTTGTAATCTCGTATGCCGTCTTCTGCTTG |
| Sequence-based reagent | Index Adapter 13 | Illumina | Jurkat 0hpi Input replicate 2 | GATCGGAAGAGCACACGTCTGAACTCCAG TCACAGTCAAATCTCGTATGCCGTCTTCTGCTTG |
| Sequence-based reagent | Index Adapter 14 | Illumina | Jurkat 3hpi Input replicate 2 | GATCGGAAGAGCACACGTCTGAACTCCAGT CACAGTTCCATCTCGTATGCCGTCTTCTGCTTG |
| Sequence-based reagent | Index Adapter 15 | Illumina | Jurkat 6hpi Input replicate 2 | GATCGGAAGAGCACACGTCTGAACTCC AGTCACATGTCAATCTCGTATGCCGTCTTCTGCTTG |
| Sequence-based reagent | Index Adapter 16 | Illumina | Jurkat 12hpi Input replicate 2 | GATCGGAAGAGCACACGTCTGAACTCCAGTC ACCCGTCCATCTCGTATGCCGTCTTCTGCTTG |
| Sequence-based reagent | Index Adapter 17 | Illumina | Jurkat 0hpi RNH-IP replicate 2 | GATCGGAAGAGCACACGTCTGAACTCCAGT CACGTAGAGATCTCGTATGCCGTCTTCTGCTTG |
| Sequence-based reagent | Index Adapter 18 | Illumina | Jurkat 3hpi RNH-IP replicate 2 | GATCGGAAGAGCACACGTCTGAACTCCAGT CACGTCCGCATCTCGTATGCCGTCTTCTGCTTG |
| Sequence-based reagent | Index Adapter 19 | Illumina | Jurkat 6hpi RNH-IP replicate 2 | GATCGGAAGAGCACACGTCTGAACTCCAG TCACGTGAAAATCTCGTATGCCGTCTTCTGCTTG |
| Sequence-based reagent | Index Adapter 20 | Illumina | Jurkat 12hpi RNH-IP replicate 2 | GATCGGAAGAGCACACGTCTGAACTCCAG TCACGTGGCCATCTCGTATGCCGTCTTCTGCTTG |
| Sequence-based reagent | Index Adapter 21 | Illumina | Jurkat 0hpi RNH +IP replicate 2 | GATCGGAAGAGCACACGTCTGAACTCCAGTC ACGTTTCGATCTCGTATGCCGTCTTCTGCTTG |
| Sequence-based reagent | Index Adapter 22 | Illumina | Jurkat 3hpi RNH +IP replicate 2 | GATCGGAAGAGCACACGTCTGAACTCCAGTC ACCGTACGATCTCGTATGCCGTCTTCTGCTTG |
| Sequence-based reagent | Index Adapter 23 | Illumina | Jurkat 6hpi RNH +IP replicate 2 | GATCGGAAGAGCACACGTCTGAACTCCAGT CACGAGTGGATCTCGTATGCCGTCTTCTGCTTG |
| Sequence-based reagent | Index Adapter 24 | Illumina | Jurkat 12hpi RNH +IP replicate 2 | GATCGGAAGAGCACACGTCTGAACTCCAGT CACGGTAGCATCTCGTATGCCGTCTTCTGCTTG |
| Sequence-based reagent | Index Adapter 25 | Illumina | CD4 0hpi Input donor 1 | GATCGGAAGAGCACACGTCTGAACTCCAGTC ACACTGATATCTCGTATGCCGTCTTCTGCTTG |
| Sequence-based reagent | Index Adapter 26 | Illumina | CD4 3hpi Input donor 1 | GATCGGAAGAGCACACGTCTGAACTCCAGTC ACATGAGCATCTCGTATGCCGTCTTCTGCTTG |
| Sequence-based reagent | Index Adapter 27 | Illumina | CD4 6hpi Input donor 1 | GATCGGAAGAGCACACGTCTGAACTCCAGTC ACATTCCTATCTCGTATGCCGTCTTCTGCTTG |
| Sequence-based reagent | Index Adapter 28 | Illumina | HeLa 0hpi Input replicate 2; CD4 12hpi Input donor 1 | GATCGGAAGAGCACACGTCTGAACTCCAGTC ACCAAAAGATCTCGTATGCCGTCTTCTGCTTG |

*Appendix 1 Continued on next page*

*Appendix 1 Continued*

| Reagent type (species) or resource | Designation | Source or reference | Identifiers | Additional information |
|---|---|---|---|---|
| Sequence-based reagent | Index Adapter 29 | Illumina | HeLa 3hpi Input replicate 2; CD4 0hpi RNH-IP donor 1 | GATCGGAAGAGCACACGTCTGAACTCCAGT CACCAACTAATCTCGTATGCCGTCTTCTGCTTG |
| Sequence-based reagent | Index Adapter 30 | Illumina | HeLa 6hpi Input replicate 2; CD4 3hpi RNH-IP donor 1 | GATCGGAAGAGCACACGTCTGAACTCCAGTC ACCACCGGATCTCGTATGCCGTCTTCTGCTTG |
| Sequence-based reagent | Index Adapter 31 | Illumina | HeLa 12hpi Input replicate 2; CD4 6hpi RNH-IP donor 1 | GATCGGAAGAGCACACGTCTGAACTCCAGTCA CCACGATATCTCGTATGCCGTCTTCTGCTTG |
| Sequence-based reagent | Index Adapter 32 | Illumina | HeLa 0hpi RNH-IP replicate 2; CD4 12hpi RNH-IP donor 1 | GATCGGAAGAGCACACGTCTGAACTCCAGT CACCACTCAATCTCGTATGCCGTCTTCTGCTTG |
| Sequence-based reagent | Index Adapter 33 | Illumina | HeLa 3hpi RNH-IP replicate 2; CD4 0hpi RNH +IP donor 1 | GATCGGAAGAGCACACGTCTGAACTCCAGT CACCAGGCGATCTCGTATGCCGTCTTCTGCTTG |
| Sequence-based reagent | Index Adapter 34 | Illumina | HeLa 6hpi RNH-IP replicate 2; CD4 3hpi RNH +IP donor 1 | GATCGGAAGAGCACACGTCTGAACTCCAGT CACCATGGCATCTCGTATGCCGTCTTCTGCTTG |
| Sequence-based reagent | Index Adapter 35 | Illumina | HeLa 12hpi RNH-IP replicate 2; CD4 6hpi RNH +IP donor 1 | GATCGGAAGAGCACACGTCTGAACTCCAGT CACCATTTTATCTCGTATGCCGTCTTCTGCTTG |
| Sequence-based reagent | Index Adapter 36 | Illumina | HeLa 0hpi RNH +IP replicate 2; CD4 12hpi RNH +IP donor 1 | GATCGGAAGAGCACACGTCTGAACTCCAG TCACCCAACAATCTCGTATGCCGTCTTCTGCTTG |
| Sequence-based reagent | Index Adapter 37 | Illumina | HeLa 3hpi RNH +IP replicate 2; CD4 0hpi Input donor 2 | GATCGGAAGAGCACACGTCTGAACTCCAGT CACCGGAATATCTCGTATGCCGTCTTCTGCTTG |
| Sequence-based reagent | Index Adapter 38 | Illumina | HeLa 6hpi RNH +IP replicate 2; CD4 3hpi Input donor 2 | GATCGGAAGAGCACACGTCTGAACTCCAGT CACCTAGCTATCTCGTATGCCGTCTTCTGCTTG |
| Sequence-based reagent | Index Adapter 39 | Illumina | HeLa 12hpi RNH +IP replicate 2; CD4 6hpi Input donor 2 | GATCGGAAGAGCACACGTCTGAACTCCAGT CACCTATACATCTCGTATGCCGTCTTCTGCTTG |
| Sequence-based reagent | Index Adapter 40 | Illumina | CD4 12hpi Input donor 2 | GATCGGAAGAGCACACGTCTGAACTCCAGT CACCTCAGAATCTCGTATGCCGTCTTCTGCTTG |
| Sequence-based reagent | Index Adapter 41 | Illumina | CD4 0hpi RNH-IP donor 2 | GATCGGAAGAGCACACGTCTGAACTCCAGT CACGACGACATCTCGTATGCCGTCTTCTGCTTG |
| Sequence-based reagent | Index Adapter 42 | Illumina | CD4 3hpi RNH-IP donor 2 | GATCGGAAGAGCACACGTCTGAACTCCAGT CACTAATCGATCTCGTATGCCGTCTTCTGCTTG |
| Sequence-based reagent | Index Adapter 43 | Illumina | CD4 6hpi RNH-IP donor 2 | GATCGGAAGAGCACACGTCTGAACTCCAG TCACTACAGCATCTCGTATGCCGTCTTCTGCTTG |
| Sequence-based reagent | Index Adapter 45 | Illumina | CD4 12hpi RNH-IP donor 2 | GATCGGAAGAGCACACGTCTGAACTCCAGT CACTCATTCATCTCGTATGCCGTCTTCTGCTTG |
| Sequence-based reagent | Index Adapter 46 | Illumina | CD4 0hpi RNH +IP donor 2 | GATCGGAAGAGCACACGTCTGAACTCCAG TCACTCCCGAATCTCGTATGCCGTCTTCTGCTTG |
| Sequence-based reagent | Index Adapter 47 | Illumina | CD4 3hpi RNH +IP donor 2 | GATCGGAAGAGCACACGTCTGAACTCCAGTC ACTCGAAGATCTCGTATGCCGTCTTCTGCTTG |
| Sequence-based reagent | Index Adapter 48 | Illumina | CD4 6hpi RNH +IP donor 2 | GATCGGAAGAGCACACGTCTGAACTCCAGTC ACTCGGCAATCTCGTATGCCGTCTTCTGCTTG |
| Sequence-based reagent | Index Adapter 49 | Illumina | CD4 12hpi RNH +IP donor 2 | GATCGGAAGAGCACACGTCTGAACTCCAGTC ACAACAACATCTCGTATGCCGTCTTCTGCTTG |
| Sequence-based reagent | AE5316 | *Achuthan et al., 2018* | First round LTR primer | TGTGACTCTGGTAACTAGAGATCCCTC |
| Sequence-based reagent | AE6380 | *Achuthan et al., 2018* | HeLa replicate 1 5dpi Linker short / HeLa replicate 1 pgR-poor DOX-Linker short / CD4 +donor 1 Linker short | TAGTCCCTTAAGCGGAG-NH2 |
| Sequence-based reagent | AE6381 | *Achuthan et al., 2018* | HeLa replicate 1 5dpi Linker long / HeLa replicate 1 pgR-poor DOX-Linker long / CD4 +donor 1 Linker long | GTAATACGACTCACTATAGGGCCTCCGCTTAAGGGAC |
| Sequence-based reagent | AE6382 | *Achuthan et al., 2018* | HeLa replicate 1 5dpi Linker primer / HeLa replicate 1 pgR-poor DOX-Linker primer / CD4 +donor 1 Linker primer | CAAGCAGAAGACGGCATACGAGATCGGTCTCGG CATTCCTGCTGAACCGCTCTTCCGATCTGTAATA CGACTCACTATAGGGC |

*Appendix 1 Continued on next page*

*Appendix 1 Continued*

| Reagent type (species) or resource | Designation | Source or reference | Identifiers | Additional information |
|---|---|---|---|---|
| Sequence-based reagent | AE6404 | *Achuthan et al., 2018* | HeLa replicate 1 5dpi Second round LTR primer DOX- / HeLa replicate 1 pgR-poor DOX- Second round LTR primer / CD4 +donor 1 Second round LTR primer | AATGATACGGCGACCACCGAGATCTACACTCTTTC CCTACACGACGCTCTTCCGATCT<u>CGATGT</u>GAGATC CCTCAGACCCTTTTAGTCAG |
| Sequence-based reagent | AE6380 | *Achuthan et al., 2018* | HeLa replicate 2 5dpi Linker short / HeLa replicate 2 pgR-poor DOX +Linker short | TAGTCCCTTAAGCGGAG-NH2 |
| Sequence-based reagent | AE6381 | *Achuthan et al., 2018* | HeLa replicate 2 5dpi Linker long /HeLa replicate 2 pgR-poor DOX +Linker long | GTAATACGACTCACTATAGGGCC TCCGCTTAAGGGAC |
| Sequence-based reagent | AE6382 | *Achuthan et al., 2018* | HeLa replicate 2 5dpi Linker primer / HeLa replicate 2 pgR-poor DOX +Linker prime | CAAGCAGAAGACGGCATACGAGATCGG TCTCGGCATTCCTGCTGAACCGCTCTT CCGATCTGTAATACGACTCACTATAGGGC |
| Sequence-based reagent | AE6404-1 | This paper | HeLa replicate 2 5dpi Second round LTR primer / HeLa replicate 2 pgR-poor DOX +Second round LTR primer | AATGATACGGCGACCACCGAGATCTACAC TCTTTCCCTACACGACGCTCTTCCGATCT<u>TTAGGC</u>G AGATCCCTCAGACCCTTTTAGTCAG |
| Sequence-based reagent | AE6386 | *Achuthan et al., 2018* | HeLa replicate 1 pgR-rich DOX-Linker short / CD4 +donor 2 Linker short | TACTATGACGGTGACGC-NH2 |
| Sequence-based reagent | AE6387 | *Achuthan et al., 2018* | HeLa replicate 1 pgR-rich DOX-Linker long / CD4 +donor 2 Linker long | GAGAATCCATGAGTATGCTCACGCGTCACCGTCATAG |
| Sequence-based reagent | AE6388 | *Achuthan et al., 2018* | HeLa replicate 1 pgR-rich DOX-Linker primer / CD4 +donor 2 Linker primer | CAAGCAGAAGACGGCATACGAGATCGGT CTCGGCATTCCTGCTGAACCGCTCTTC CGATCTGAGAATCCATGAGTATGCTCAC |
| Sequence-based reagent | AE6406 | *Achuthan et al., 2018* | HeLa replicate 1 pgR-rich DOX-Second round LTR primer / CD4 +donor 2 Second round LTR primer | AATGATACGGCGACCACCGAGATCTACA CTCTTTCCCTACACGACGCTCTTCCGATCT<u>ACAGTG</u>GA GATCCCTCAGACCCTTTTAGTCAG |
| Sequence-based reagent | AE6456 | *Achuthan et al., 2018* | HeLa replicate 1 pgR-poor DOX +Linker short | TAGACTGACGCAGTCTG-NH2 |
| Sequence-based reagent | AE6457 | *Achuthan et al., 2018* | HeLa replicate 1 pgR-poor DOX +Linker long | GACGTACATACTGATCGCATAGCAGACTGCGTCAGTC |
| Sequence-based reagent | AE6458 | *Achuthan et al., 2018* | HeLa replicate 1 pgR-poor DOX +Linker primer | CAAGCAGAAGACGGCATACGAGATCGGT CTCGGCATTCCTGCTGAACCGCTCTTCCG ATCTGACGTACATACTGATCGCATAG |
| Sequence-based reagent | AE6405 | *Achuthan et al., 2018* | HeLa replicate 1 pgR-poor DOX +Second round LTR primer | AATGATACGGCGACCACCGAGATCTAC ACTCTTTCCCTACACGACGCTCTTCCGA TCT<u>TGACCA</u>GAGATCCCTCAGACCCTTTTAGTCAG |
| Sequence-based reagent | AE6386 | *Achuthan et al., 2018* | HeLa replicate 2 pgR-rich DOX +Linker short | TACTATGACGGTGACGC-NH2 |
| Sequence-based reagent | AE6387 | *Achuthan et al., 2018* | HeLa replicate 2 pgR-rich DOX +Linker long | GAGAATCCATGAGTATGCTCACGCGTCACCGTCATAG |
| Sequence-based reagent | AE6388 | *Achuthan et al., 2018* | HeLa replicate 2 pgR-rich DOX +Linker primer | CAAGCAGAAGACGGCATACGAGATCG GTCTCGGCATTCCTGCTGAACCGCTCTTC CGATCTGAGAATCCATGAGTATGCTCAC |
| Sequence-based reagent | AE6406-1 | This paper | HeLa replicate 2 pgR-rich DOX +Second round LTR primer | AATGATACGGCGACCACCGAGATCTACAC TCTTTCCCTACACGACGCTCTTCCGATC T<u>GCCAAT</u>GAGATCCCTCAGACCCTTTTAGTCAG |
| Sequence-based reagent | AE6456 | *Achuthan et al., 2018* | HeLa replicate 3 pgR-rich DOX-Linker short | TAGACTGACGCAGTCTG-NH2 |
| Sequence-based reagent | AE6457 | *Achuthan et al., 2018* | HeLa replicate 3 pgR-rich DOX-Linker long | GACGTACATACTGATCGCATAGCAGACTGCGTCAGTC |
| Sequence-based reagent | AE6458 | *Achuthan et al., 2018* | HeLa replicate 3 pgR-rich DOX-Linker primer | CAAGCAGAAGACGGCATACGAGATCGGT CTCGGCATTCCTGCTGAACCGCTCTTCC GATCTGACGTACATACTGATCGCATAG |
| Sequence-based reagent | AE6411 | *Achuthan et al., 2018* | HeLa replicate 3 pgR-rich DOX-Second round LTR primer | AATGATACGGCGACCACCGAGATCTA CACTCTTTCCCTACACGACGCTCTTCC GATCT<u>AGTTCC</u>GAGATCCCTCAGACCCTTTTAGTCAG |

*Appendix 1 Continued on next page*

*Appendix 1 Continued*

| Reagent type (species) or resource | Designation | Source or reference | Identifiers | Additional information |
|---|---|---|---|---|
| Sequence-based reagent | R-loop oligo1* | *Kang et al., 2021* | R-loop; R:D+ssDNA | 5'-[Cy3]-GCC AGG GAC GAG GTG AAC CTG CAG GTG GGC GGC TAC TAC TTA GAT GTC ATC CGA GGC TTA TTG GTA GAA TTC GGC AGC GTC ATG C GA CGG C-3' |
| Sequence-based reagent | R-loop oligo2* | *Kang et al., 2021* | R-loop; dsDNA | 5'-GCC GTC GCA TGA CGC TGC CGA ATT CTA CCA CGC GAT TCA TAC CTG TCG TGC CAG CTG CTT TGC CCA CCT GCA GGT TCA CCT CGT CCC TGG C-3' |
| Sequence-based reagent | R-loop RNA | *Kang et al., 2021* | R-loop; R:D+ssDNA | 5'-[Cy5]-GCA GCU GGC ACG ACA GGU AUG AAU C-3' |
| Sequence-based reagent | Homoduplex | *Kang et al., 2021* | dsDNA | 5'-[Cy3]-GCC AGG GAC GAG GTG AAC CTG CAG GTG GGC AAA GCA GCT GGC ACG ACA GGT ATG AAT CGC GTG GTA GAA TTC GGC AGC GTC ATG CGA CGG C-3' |
| Sequence-based reagent | Hybrid DNA | *Kang et al., 2021* | Hybrid | 5'-CCC ATA CCG TAT AAC CAT TTG GCT GTC AAG CT CCG GGT-3' |
| Sequence-based reagent | Hybrid RNA | *Kang et al., 2021* | Hybrid | 5'-[Cy5]-ACC CGG AGC UU G GAC AGC CAA AUG GU U AUA CGG UAU GGG-3' |
| Sequence-based reagent | oligo 5 | *Kang et al., 2021* | R:D+ssDNA | 5'GCAGTAGCATGACGCTGCTG AATTCTACCACGCTATGCT CTCGTCTA GGTTCACTCCGT CCCTGCGATTCATACCTGTCGTGCCAGCTGC |
| Other | Jurkat integration site | *Li et al., 2020b* | SRR12322252 | Source data |
| Other | TSA-seq_SPAD | *Chen et al., 2018* | SRR3538917, SRR3538918, SRR3538919, SRR3538920 | Source data |
| Other | SPIN (Spatial Position Inference of the Nuclear genome) annotation of speckle | *Wang et al., 2020* | https://github.com/ma-compbio/SPIN; *ma-compbio, 2020* | Source data |
| Other | LADs | *Meuleman et al., 2013* | GSE22428 | Source data |

