## [Editor Report · eLife Assessment]

This study presents two main findings regarding HIV-1 genomic integration. The first, based on **convincing** evidence in primary cell models, is that HIV-1 induces R loop formation, though the viral driver of this process remains undefined. The second, based on model cell systems with limited physiological relevance to HIV-1, is that a portion of HIV-1 genomes integrates in the vicinity of where R loops form. This finding has the potential to offer **fundamental** insight into HIV-1 integration, but the strength of the presented evidence was viewed as incomplete and needing additional validation by more direct experimental methods in order to understand what the mechanistic relationship between the formation of R loops and HIV-1 integration is.

---

## [Referee Report · Reviewer #1 (Public review)]

(1) Significance of findings and strength of evidence.

(a) The work presented in this manuscript is intended to support the authors' novel idea that HIV DNA integration strongly favors "triple-stranded" R-loops in DNA formed either during transcription of many, but not all, genes or by strand invasion of silent DNA by transcripts made elsewhere, and that HIV infection promotes R-loop formation mediated by incoming virions in the absence of reverse transcription. The authors were able to demonstrate a reverse transcription-independent increase in R-loop formation early during HIV infection, while also demonstrating increased integration into sequences that contain R-loop structures. Furthermore, this manuscript also identifies that R-loops are present in both transcriptional active and silent regions of the genome and that HIV integrase interacts with R-loops. Although the work presented supports a correlation between R-loop formation and HIV DNA integration, it does not prove the authors' hypothesis that R-loops are directly targeted for integration. Direct experimentation, such as in vitro integration into defined DNA targets, will be required. Further, the authors provide no explanation as to how current sophisticated structural models of concerted retroviral DNA integration into both strands of double-stranded DNA targets can accommodate triple-stranded structures. Finally, there are serious technical concerns with interpretation of the integration site analyses.

This resubmitted manuscript has corrected some of the issues raised by the previous reviews - particularly the quality of the English - but otherwise the text and figures remain very much the same and concerns regarding the conclusions drawn regarding integration site specificity remain. The manuscript also still suffers from a lack of description of experimental detail necessary to understand the results as presented. In many cases, explanations given privately in the rebuttal o the earlier reviews need to be made available to all readers, not just the reviewers.

(2) Public review with guidance for readers around how to interpret the work, highlighting important findings but also mentioning caveats.

(a) Introduction: The authors provide an excellent introduction to R-loops but they base the rationale for this study on mis-citation of earlier studies regarding integration in transcriptionally silent regions of the genome. The "most favored locus" cited in the very old reference 6 comprises only 5 events and has not been reproduced in more recent, much larger datasets For example, see the study of over 300.000 sites in ref 14. The laundry list of IN interactors in lines 43-44 is based on old experiments. It is now quite clear that the only direct interaction of importance is with LEDGF and that should be discussed here. Also discussed should be the role of the capsid in the nuclear entry and targeting. For example, one of the references cited, as well as a mention in the discussion (Line 326) concerns CPSF-6, which is now known to modulate nuclear entry and specificity by interacting with capsid, not integrase. The statement on lines 46-47 regarding that some highly expressed genes are, nonetheless, poor targets for integration is correct, but the experiment cited was done in PBMC with wild-type HIV-1and it is possible that those genes were expressed in non-target cells like B-cells or monocytes.

(b) Figure 1: Demonstrates models for HIV infections in both cell lines and primary human CD4+ T cells. R-loop formation was determined through a method called DRIPc-seq which utilizes an anti-body specific for DNA-RNA hybrid structures and sequences these regions of the genome using RNaseH treatment to show that when RNA-DNA hybrids are absent then no R-loops are detected. In these models of in vitro and ex vivo infection, the authors show that R-Loop formation increases following HIV infection between 6 hr. post-infection and 12 hrs. post-infection, depending on the cell model. However, these figures lack a mock infected control for each cell model to assess R-loop formation at the same time points. They would also benefit from a control showing that virus entry is necessary, such as omitting the VSV G protein donor.

(c) Figure 2: This figure shows that cells infected with HIV show more R-loops as well as longer sequences containing R-loop structures. Panel B shows that these R-loops were distributed throughout different genomic features, such as both genic and intergenic regions of the genome. However, the data are presented in such a way that it is impossible to determine the proportion of R-loops in each type of genomic feature. The reader has no way to tell, for example, the proportion of R-loops in genic vs intergenic DNA and how this value changes with time. Furthermore, increased R-loop formation due to HIV infection showed poor correlation with gene expression, suggesting that R-loops were not forming due to transcriptional activation, although the difference between 0 and the remaining timepoints is not apparent, nor is the meaning of the absurd p values.

The experiments presented in Figures 1 and 2 show that treatment of cells with VSV G-pseudotyped HIV-1 leads to a significant increase in R loops in all parts of the genome. Accumulation of R-loops at so soon after infection, as well as its resistance to RT and Integration inhibitors, rules out the involvement of newly synthesized viral DNA or any newly made viral protein (Figure S3). Rather, some component(s) of the virion, possibly protease, or an accessory gene product such as Vpr or Vif, must be directly responsible e (although the authors neglect to draw this conclusion in the description of these experiments, lines 125-135, leaving it hanging until the Discussion).

On the whole, and as a non-expert in this area, I find the overall conclusions of this part of the study convincing, but, as pointed out in one of the earlier reviews, the virologic significance of early effects seen at high multiplicity of infection (likely hundreds of particles per cell) needs to be taken with a grain of salt. At a minimum, this point should be discussed. Also, the study would have been greatly strengthened by a simple experiment to identify the virion protein responsible for the effect.

Based on the results in the first two figures, the authors hypothesize that R-Loop induction early in infection plays an important role in HIV replication, specifically by interacting with the intasome and thus directing integration to regions of the host genome favorable for expression of the provirus. Experiments to test this idea and probe the mechanism are described in the remaining 3 figures, which, despite comments in the previous reviews, are unchanged from the previous version and still suffer from serious defects in experimental design and interpretation.

(d) Figure 3: This figure shows the use of cell lines carrying R-loop inducible (mAIRN) or non-inducible (ECFP) genes to model association of HIV integration with R-loop structures. The authors demonstrate the functional validation of R-loop induction in the cell line model. Additionally, when R-loops are induced there is a significant increase in HIV integration in the R-loop forming vector sequence when R-loops are induced with doxycycline. This result shows a correlation between expression and integration that is much stronger in the R-loop forming gene than in the unreferenced ECFP gene but does not prove that integration directly targets R-loops. It is possible, for example, that some feature of the DNA sequence, such as base composition affects both integration and R-loop formation independently. As described more fully below, there is also a serious concern regarding the method used to quantitate the integration frequencies. As before, There are a number of problems here.

(1) The authors use a classic, but suboptimal integration site assay comprising restriction enzyme digestion followed by PCR to assess integration site distribution, and (despite statements to the contrary in the rebuttal) read counts to quantitate relative frequencies of target site use. See the legend and axis labels in Fig 3E, F, and G. This approach leads to serious bias in the ability to detect and count the use of integration sites that are either too close or too far from the sites of cleavage and can lead to artefactual misrepresentation of their chromosomal distribution.

(2) The result shown in Figure 3D is uninterpretable. It is simply not possible that the 3-fold increase in luciferase activity is due addition of 25 10-kb sequences leading to A 3-fold increase in integration frequency into the target sequence, particularly when panel E shows that the measured frequency is on the order of 20 reads per million. Something else must be going on here.

(3) Panels 3F and G show the read count distribution in the introduced target sequences plotted in a completely nonstandard way and is explained so poorly that I could not be sure what the authors were trying to show. The numbers on the bottom of the 2 plots appear to represent the only sites of integration seen in the 10-kb region studied. If so, this is not the expected result for the authors claim of greatly increasing regional integration. As can easily be seen in the figures of ref 14, high frequency gene targets are characterized by large numbers of sites, not by more frequent targeting of small numbers of sites as implied by the figures.

(e) Figure 4: This figure shows evidence of increased HIV integration within regions of the genome containing R-loops with additional preference with integration within the R-loop and decrease in frequency of integration further from the R-loop. Identifying a preference for R-loops is very intriguing but the authors do also demonstrate that integration does occur when R-loops are not present. Also Panel A, which shows that regions of cell DNA that form R-loops have a higher frequency of Integration sites than those that do not, should also be controlled for the level of gene expression of the two types of region. the result shown cannot be interpreted to mean that R-loops have anything to do with integration targeting. It is already well-established that about 80% of HIV integration sites are in expressed genes, which comprise about 20% of the genome. Since a gene must be expressed to contain an R-loop, the non-R-loop fraction will contain the 80% of the genome that is a 20-fold poorer target, giving the result shown, whether R-loops are involved or not. The rather weak correlation between R-Loop locations and integration site distribution in Fig 4C and D hardly seems consistent with the curves seen in 4B. Can the authors refute the hypothesis that the apparent correlation is simply because both integration and R-Loop formation frequency must correlate with level of gene expression and therefore their correlation with one another cannot be used to infer causality/ As pointed out in prior reviews, R-loops themselves cannot be targets for integration. In their rebuttal, the authors agree and have made slight modifications to their conclusion in the text, now concluding that Integration favors the vicinity of an R-loop. Why then do the peaks in correlation curves in Fig 4B center exactly on the center of the R-loops? It seems that this result would be more consistent with integration and R-loop formation favoring the same sites, but for different reasons (base composition for example).

(f) Figure 5: In this figure the authors demonstrate that HIV integrase binds to R-loops through a number of protein assays, but does not show that this binding is associated with enzymatic activity. EMSA of integrase identified increased binding to DNA-RNA over dsDNA. Additionally, precipitation of RNA-DNA hybrids pulled down HIV integrase. A proximity ligation assay detecting R-loops and HIV-integrase showed co-localization within the nucleus of HeLa cells. HeLa cells were probably used due to their efficiency of transduction but are not physiologically relevant cell types. Figure 5 suffers greatly in interpretability from the failure of the authors to use assembled intasomes, since the DNA binding properties are likely to be quite different. The authors excuse that they were unable to prepare intasomes (which needs to be included in the text, not just in the rebuttal) explains but does not justify the use of monomeric IN protein. Figure 5A shows that the IN binding is NOT specific to R-loops, since any single-stranded DNA binds equally. The authors should make this point in the text.

The experiment using integrase overexpression in cells brings up some déjà vu to a retrovirologist. There is some history in retrovirology of experiments like this having been used to draw conclusions (like the role of integrase in nuclear import) that have since proven to be wrong. Also, Fig 5G is not interpretable quantitively, since the distribution of neither IN nor R-loops is probed, and we have no idea what proportion of each is in the PLA spots. Overall, this section would be much more convincing if it also included some direct experimentation, such as in vitro integration using intasomes, or infection of cells with viral mutants (or in the presence of inhibitors) affecting the function of whatever virion protein found to be important for R-loop formation.

(g) Discussion: In the discussion, the authors address how their work relates to previous evidence of HIV integration by association of LEDGF/p75 and CPSF6. They also cite that LEDGF/p75 has possible R-loop binding capabilities. They also discuss what possible mechanisms are driving increases in R-loop formation during HIV infection, pointing to possible HIV accessory proteins. They also state that how HIV integrates in transcriptionally silent regions is still unknown but do point out that they were able to show R-loops appear in many different regions of the genome but did not show that R-loops in transcriptional inactive regions are integration targets. More seriously, they failed to make a connection between their work and current understanding of the biochemical and structural mechanism of the integration reaction.

---

## [Referee Report · Reviewer #3 (Public review)]

In this manuscript, Park and colleagues describe a series of experiments that investigate the role of R-loops in HIV-1 genome integration. The authors show that during HIV-1 infection, R-loops levels on the host genome accumulate. Using a synthetic R-loop prone gene construct, they show that HIV-1 integration sites target sites with high R-loop levels. They further show that integration sites on the endogenous host genome are correlated with sites prone to R-loops. Using biochemical approaches, as well as in vivo co-IP and proximity ligation experiments, the authors show that HIV-1 integrase physically interacts with R-loop structures.

The major strengths of this work is that the investigators use multiple independent experimental systems and multiple cell types to support their conclusions, including in vivo and biochemical experiments. Furthermore, their use of genome-wide analyses help to support their conclusion that HIV targets genomic regions enriched with R-loops versus those lacking such enrichment.

This work may have a significant impact on the field of HIV genomic integration by elucidating why transcription levels are not the sole determinant of HIV integration sites.

---

## [Author Response]

The following is the authors’ response to the original reviews.

**Reviewer #1 (Public Review):**
We appreciate the valuable and constructive comments of Reviewer #1 on our manuscript. We have addressed the comments from Reviewer #1 in the public review in the response to the recommendations for the authors, as the public review comments largely overlap with that of the recommendations for the authors.
**Reviewer #1 (Recommendations For The Authors):**
(1.1) Figure 1 did not use a mock-infected control for the development of R-loops but only a time before infection. I think it would have been a good control to have that after the same time of infection non-infected cells did not show increases in R-loops and this is not a product of the cell cycle.

We prepared our DRIPc-seq library using cell extracts harvested at 0, 3, 6, and 12 h post-infection (hpi), all at the same post-seeding time point. Each sample was infected with HIV-1 virus in a time-dependent manner. Therefore, it is unlikely that the host cellular R-loop induction observed in our DRIPc-seq results was due to R-loop formation during the cell cycle. In Lines 93–95 of the Results section of the revised manuscript, we have provided a more detailed description of our DRIPc-seq library experimental scheme. Thank you.

(1.2) Figure 2 should have included a figure showing the proportion of DRIPc-seq peaks located in different genome features relative to one another instead of whether they were influenced by time post-infection. Figure 2C was performed in HeLa cells, but primary T cell data would have been more relevant as primary CD4+ T cells are more relevant to HIV infection.

We have included a new figure presenting the relative proportion of DRIPc-seq peaks mapped to different genomic features at each hpi (Fig. 2C of the revised manuscript). We found that the proportion of DRIPc-seq peaks mapped to various genomic compartments remained consistent over the hours following the HIV-1 infection. This further supports our original claim that HIV-1 infection does not induce R-loop enrichment at specific genomic features but that the accumulation of R-loops after HIV-1 infection is widely distributed.

We considered HeLa cells as the primary in vitro infection model, therefore, we conducted RNA-seq only on HeLa cells. However, we agree with the reviewer's opinion that data from primary CD4+ T cells may be more physiologically relevant. Nevertheless, as demonstrated in the new figure (Fig. 2C of the revised manuscript), HIV-1 infection did not significantly alter the proportion of R-loop peaks mapped to specific genomic compartments, such as gene body regions, in HeLa, primary CD4+ T, and Jurkat cells. Therefore, we anticipate no clear correlation between changes in gene expression levels and R-loop peak detection upon HIV-1 infection, even in primary T cells. Thank you.

(1.3) Figure 5G is very hard to see when printed, is there a change in brightness or contrast that could be used? The arrows are helpful but they don't seem to be pointing to much.

We have highlighted the intensity of the PLA foci and magnified the images in Fig. 5G in the revised manuscript. While editing the images according to your suggestion, we found a misannotation regarding the multiplicity of infection in the number of PLA foci per nucleus quantification analysis graph in Fig. 5G of the original manuscript. We have corrected this issue and hope that it is now much clearer.

(1.4) The introduction provided a good background for those who may not have a comprehensive understanding of DNA-RNA hybrids and R-loops, but the rationale that integration in non-expressed sequence implies that R-loops may be involved is very weak and was not addressed experimentally. A better rationale would have been to point out that, although integration in genes is strongly associated with gene expression, the association is not perfect, particularly in that some highly expressed genes are, nonetheless, poor integration targets.

In accordance with the reviewer's comment, we revised the Introduction. We have deleted the statement and reference in the introduction "... the most favored region of HIV-1 integration is an intergenic locus, ...”, which may overstate the relevance of the R-loop in HIV-1 integration events in non-expressed sequences. Instead, we introduced a more recent finding that high levels of gene expression do not always predict high levels of integration, together with the corresponding citation (Lines 46– 47 of the revised manuscript), according to the reviewer’s suggestion in the reviewer's public review 2-(a).

(1.5) The discussion was seriously lacking in connecting their conclusions regarding R-loop targeting of integration to how integration works at the structural level, where it is very clear that concerted integration on the two DNA strands ca 5 bp apart is essential to correct, 2-ended integration. It is very difficult to visualize how this would be possible with the triple-stranded R-loop as a target. The manuscript would be greatly strengthened by an experiment showing concerted integration into a triplestranded structure in vitro using PICs or pure integrase.

We believe there has been a misunderstanding of our interpretation regarding the putative role of R-loop structures in the HIV-1 integration site mechanism because of some misleading statements in our original manuscript. Based primarily on our current data, we believe that R-loop structures are bound by HIV-1 integrase proteins and lead to HIV-1 viral genome integration into the vicinity regions of the host genomic R-loops. By carefully revising our manuscript, we found that the title, abstract, and discussion of our original manuscript includes phrases, such as “HIV-1 targets R-loops for integration,” which may overstate our finding on the role of R-loop in HIV-1 integration site selection. We replaced these phrases. For example, we used phrases, such as, “HIV-1 favors vicinity regions of R-loop for the viral genome integration,” in the revised manuscript. We apologize for the inconvenience caused by the unclear and nonspecific details of our findings.

Using multiple biochemical experiments, we successfully demonstrated the interaction between the cellular R-loop and HIV-1 integrase proteins in cells and in vitro (Fig. 5 of the revised manuscript). However, we could not validate whether the center of the triple-stranded R-loops is the extraction site of HIV-1 integration, where the strand transfer reaction by integrase occurs. This is because an R-loop can be multi-kilobase in size (1, 2); therefore, we displayed a large-scale genomic region (30-kb windows) to present the integration sites surrounding the R-loop centers. Nevertheless, we believe that we validated R-loop-mediated HIV-1 integration in R-loop-forming regions using our pgR-poor and pgR-rich cell line models. When infected with HIV-1, pgR-rich cells, but not pgR-poor cells, showed higher infectivity upon R-loop induction in designated regions following DOX treatment (Fig. 3C and 3D of the revised manuscript). In addition, we quantified site-specific integration events in R-loop regions, and found that a greater number of integration events occurred in designated regions of the pgR-rich cellular genome upon R-loop induction by DOX treatment, but not in pgR-poor cells (Fig. 3E–G of the revised manuscript).

We agree with the reviewer that an experiment showing the concerted integration of purified PICs into a triple-stranded structure in vitro would greatly strengthen our manuscript. We attempted the purification of viral DNA (vDNA)-bound PICs using either Sso7d-tagged HIV-1 integrase proteins or non-tagged HIV-1 integrase proteins (F185K/C280S) procured from the NIH HIV reagent program (HRP-20203), following the method described by Passos et al., Science, 2017; 355 (89-92) (3). Despite multiple attempts, we could not purify the nucleic acid-bound protein complexes for in vitro integration assays. However, we believe that pgR-poor and pgR-rich cell line models provide a strong advantage in specificity of our primer readouts. Compounded with our in cellulo observation, we believe that our work provides strong evidence for a causative relationship between R-loop formation/R-loop sites and HIV-1 integration.

Additionally, in the Discussion section of the revised manuscript, we have expanded our discussion on the role of genomic R-loops contributing in molding the host genomic environment for HIV-1 integration site selection, and the potential explanation on how R-loops are driving integration over long-range genomic regions. Thank you.

(1.6) There are serious concerns with the quantitation of integration sites used here, which should be described in detail following line 503 but isn't. In Figure 3, E-G, they are apparently shown as reads per million, while in Figure 4B as "sites (%)" and in 4C as log10 integration frequency." Assuming the authors mean what they say, they are using the worst possible method for quantitation. Counting reads from restriction enzyme-digested, PCR-digested DNA can only mislead. At the numbers provided (MOI 0.6, 10 µg DNA assayed) there would be about 1 million proviruses in the samples assayed, so the probability of any specific site being used more than once is very low, and even less when one considers that a 10% assay efficiency is typical of integration site assays. Although the authors may obtain millions of reads per experiment, the number of reads per site is an irrelevant value, determined only by technical artefacts in the PCR reactions, most significantly the length of the amplicons, a function of the distance from the integration site to the nearest MstII site, further modified by differences in Tm. Better is to collapse identical reads to 1 per site, as may have been done in Figure 4B, however, the efficiency of integration site detection will still be inversely related to the length of the amplicon. Indeed, if the authors were to plot the read frequency against distance to the nearest MstII site, it is likely that they would get plots much like those in Figure 4B.

Detailed methods for integration site sequencing data processing are described in the Materials and Methods section of the revised manuscript (Line 621–631 of the revised manuscript). We primarily followed HIV-1 integration site sequencing data processing methods previously described by Li et al., mBio, 2020; 11(5) (4).

While it may be correct that the HIV-1 integration event cannot occur more than once at a given site, our Fig. 3E, 4C, and 4D of the revised manuscript present the number of integration-site sequencing read counts expressed in reads-per-million (RPM) units or as log10-normalized values. Based on the number of mapped reads from the integration site sequencing results, we can infer that there was an integration event at this site, whether it was a single or multiple event.

We believe that the original annotation of y-axis, “Integration frequency,” may be misleading as it can be interpreted as a probability of any specific site being used for HIV-1 integration. Therefore, we corrected it as “number of mapped read” for clarity (Fig. 3E–G, 4C and 4D, and the corresponding figure legends of the revised manuscript). We apologize for any confusion. Thank you.

Other points:(1.7) Overall: There are numerous grammatical and usage errors, especially in agreement of subject and verb, and missing articles, sometimes multiple times in the same sentence. These must be corrected prior to resubmission.

The revised manuscript was edited by a professional editing service. Thank you.

(1.8) Line 126-134: A striking result, but it needs more controls, as discussed above, including a dose-response analysis.

We determined the doses of NVP and RAL inhibitors in HeLa cells by optimizing the minimum dose of drug treatment that provided a sufficient inhibitory effect on HIV1 infection (Author response image 1). The primary objective of this experiment was to determine R-loop formation while reverse transcription or integration of the HIV-1 life cycle was blocked, therefore, we do not think that a dose-dependent analysis of inhibitors is required.

**Author response image 1. sa3fig1:** Representative flow cytometry histograms of VSV-G-pseudotyped HIV-1-EGFP-infected HeLa cells at an MOI of 1, harvested at 48 hpi. The cells were treated with DMSO, the indicated doses of nevirapine (NVP) (A) or indicated doses of raltegravir (RAL) (B) for 24 h before infection.

(1.9) Line 183: Please tell us what ECFP is and why it was chosen. Is there a reference for its failure to form R-loops?Ibid: The human AIRN gene is a very poor target for HIV integration in PBMC.

A high GC skew value (> 0) is a predisposing factor for R-loop formation at the transcription site. This is because a high GC skew causes a newly synthesized RNA strand to hybridize to the template DNA strand, and the non-template DNA strand remains looped out in a single-stranded conformation (5) (Ref 36 in the revised manuscript). The ECFP sequence possessed a low GC skew value, as previously used for an R-loop-forming negative sequence (6) (Ref 17 of the revised manuscript). We have added this description and the corresponding references to Lines 188–192 of the revised manuscript.

The human AIRN gene (RefSeq DNA sequence: NC_000006.12) sequence possesses a GC skew value of -0.04, in a window centered at base 2186, while the mouse AIRN (mAIRN) sequence is characterized by a GC skew value of 0.213. The ECFP sequence gave a GC skew value of -0.086 in our calculation. We anticipated that the human AIRN gene region does not form a stable R-loop, and in fact, it did not harbor R-loop enrichment upon HIV-1 infection in our DRIPc-seq data analysis of multiple cell types (Author response image 2)

**Author response image 2. sa3fig2:** Genome browser screenshot over the chromosomal regions in 20-kb windows centered on human AIRN showing results from DRIPc-seq in the indicated HIV-1-infected cells (blue, 0 hpi; yellow, 3 hpi; green, 6 hpi; red, 12 hpi).

(1.10) Line 190: You haven't shown dependence. Associated is a better word.

Thank you for the suggestion. We have changed “R-loop-dependent site-specific HIV-1 integration events...” to “R-loop-associated site-specific HIV-1 integration events...” (Line 198 of the revised manuscript) according to the reviewer’s suggestion in the revised manuscript.

(1.11) Line 239: What happened to P1? What is the relationship of the P and N regions to genes?

We have added superimpositions of the P1 chromatin region on DRIPc-seq and the HIV-1 integration frequency to Figure 4C of the revised manuscript. We observed a relevant integration event within the P1 R-loop region, but to a lesser extent than in the P2 and P3 R-loop regions, perhaps because the P1 region has relatively less R-loop enrichment than the P2 and P3 regions, as examined by DRIP-qPCR in S3A Fig. of the revised manuscript.

Genome browser screenshots with annotations of accommodating genes in the P and N regions are shown in S2A–E Fig. of the revised manuscript, and RNA-seq analysis of the relative gene expression levels of the P1-3 and N1,2 R-loop regions are shown in S4 Table of the revised manuscript. Thank you.

(1.12) Line 261: But the binding affinity of integrase to the R-loop is somewhat weaker than to double-stranded DNA according to Figure 5A.

Nucleic acid substrates were loaded at the same molarity, and the percentage of the unbound fraction was calculated by dividing the intensity of the unbound fraction in each lane by the intensity of the unbound fraction in the lane with 0 nM integrase in the binding reaction. The calculated percentages of the unbound fraction from three independent replicate experiments are shown in Fig. 5A, right of the revised manuscript. In our analysis and measurements, the integrase proteins showed higher binding affinities to the R-loop and R-loop comprising nucleic acid structures than to dsDNA in vitro. We hope that this explanation clarifies this point.

(1.13) Line 337: "accumulate". This is a not uncommon misinterpretation of the results of studies on the distribution of intact proviruses in elite controllers. The only possible correct interpretation of the finding is that proviruses form everywhere else but cells containing them are eliminated, most likely by the immune system.

Thank you for the suggestion. We have changed the Line 337 of the original manuscript to “... HIV-1 proviruses in heterochromatic regions are not eliminated but selected by immune system,” in Lines 361-363 of the revised manuscript.

(1.14) Line 371 How many virus particles per cell does this inoculum amount to?

We determined the amount of GFP reporter viruses required to transduce ∼50% of WT Jurkat T cells, corresponding to an approximate MOI of 0.6. We repeatedly obtained 30–50% of VSV-G-pseudotyped HIV-1-EGFP positively infected cells for HIV1 integration site sequencing library construction for Jurkat T cells.

(1.15) Line 503 and Figures 3 and 4: There must be a clear description of how integration events are quantitated.

Detailed methods for integration site sequencing data processing are described in the Materials and Methods section of the revised manuscript (Line 621–631 of the revised manuscript). We primarily followed HIV-1 integration site sequencing data processing methods previously described in Li et al., mBio, 2020; 11(5) (4).

**Reviewer #2 (Public Review):**
Retroviral integration in general, and HIV integration in particular, takes place in dsDNA, not in R-loops. Although HIV integration can occur in vitro on naked dsDNA, there is good evidence that, in an infected cell, integration occurs on DNA that is associated with nucleosomes. This review will be presented in two parts. First, a summary will be provided giving some of the reasons to be confident that integration occurs on dsDNA on nucleosomes. The second part will point out some of the obvious problems with the experimental data that are presented in the manuscript.

We appreciate your comments. We have carefully addressed the concerns expressed as follows (your comments are in italics):

(2.1) 2017 Dos Passos Science paper describes the structure of the HIV intasome. The structure makes it clear that the target for integration is dsDNA, not an R-loop, and there are very good reasons to think that structure is physiologically relevant. For example, there is data from the Cherepanov, Engelman, and Lyumkis labs to show that the HIV intasome is quite similar in its overall structure and organization to the structures of the intasomes of other retroviruses. Importantly, these structures explain the way integration creates a small duplication of the host sequences at the integration site. How do the authors propose that an R-loop can replace the dsDNA that was seen in these intasome structures?

We do appreciate the current understanding of the HIV-1 integration site selection mechanism and the known structure of the dsDNA-bound intasome. Our study proposes an R-loop as another contributor to HIV-1 integration site selection. Recent studies providing new perspectives on HIV-1 integration site targeting motivated our current work. For instance, Ajoge et al., 2022 (7) indicated that a guanine-quadruplex (G4) structure formed in the non-template DNA strand of the R-loop influences HIV-1 integration site targeting. Additionally, I. K. Jozwik et al., 2022 (8) showed retroviral integrase protein structure bound to B-to-A transition in target DNA. R-loop structures are a prevalent class of alternative non-B DNA structures (9). We acknowledge the current understanding of HIV-1 integration site selection and explore how R-loop interactions may contribute to this knowledge in the Discussion section of our manuscript.

Primarily based on our current data, we believe that R-loop structures are bound by HIV-1 integrase proteins and lead to HIV-1 viral genome integration into the vicinity regions of the host genomic R-loops, but we do not claim that R-loops completely replace dsDNA as the target for HIV-1 integration. An R-loop can be multi-kilobase in size and the R-loop peak length widely varies depending on the immunoprecipitation and library construction methods (1, 2), therefore, we could not validate whether the center of triple-stranded R-loops is the extraction site of HIV-1 integration where the strand transfer reaction by integrase occurs. Therefore, we replaced phrases such as, “HIV-1 targets R-loops for integration,” which may overstate our finding on the role of R-loop in HIV-1 integration site selection, with phrases, such as, “HIV-1 favors vicinity regions of R-loop for the viral genome integration,” in the revised manuscript. We apologize for the inconvenience caused by the unclear and non-specific details of our findings. Nevertheless, we believe that we validated R-loop-mediated HIV-1 integration in R-loop-forming regions using our pgR-poor and pgR-rich cell line models. We quantified site-specific integration events in the R-loop regions, and found that a greater number of integration events occurred in designated regions of the pgR-rich cellular genome upon R-loop induction by DOX treatment, but not in pgR-poor cells (Fig. 3E–G of the revised manuscript).

dsDNA may have been the sole target of the intasome demonstrated in vitro possibly because dsDNA has only been considered as a substrate for in vitro intasome assembly. We hope that our work will initiate and advance future investigations on target-bound intasome structures by considering R-loops as potential new targets for integrated proteins and intasomes.

(2.2) As noted above, concerted (two-ended) integration can occur in vitro on a naked dsDNA substrate. However, there is compelling evidence that, in cells, integration preferentially occurs on nucleosomes. Nucleosomes are not found in R loops. In an infected cell, the viral RNA genome of HIV is converted into DNA within the capsid/core which transits the nuclear pore before reverse transcription has been completed. Integration requires the uncoating of the capsid/core, which is linked to the completion of viral DNA synthesis in the nucleus. Two host factors are known to strongly influence integration site selection, CPSF6 and LEDGF. CPSF6 is involved in helping the capsid/core transit the nuclear pore and associate with nuclear speckles. LEDGF is involved in helping the preintegration complex (PIC) find an integration site after it has been released from the capsid/core, most commonly in the bodies of highly expressed genes. In the absence of an interaction of CPSF6 with the core, integration occurs primarily in the lamin-associated domains (LADs). Genes in LADs are usually not expressed or are expressed at low levels. Depending on the cell type, integration in the absence of CPSF6 can be less efficient than normal integration, but that could well be due to a lack of LEDGF (which is associated with expressed genes) in the LADs. In the absence of an interaction of IN with LEDGF (and in cells with low levels of HRP2) integration is less efficient and the obvious preference for integration in highly expressed genes is reduced. Importantly, LEDGF is known to bind histone marks, and will therefore be preferentially associated with nucleosomes, not R-loops. LEDGF fusions, in which the chromatin binding portion of the protein is replaced, can be used to redirect where HIV integrates, and that technique has been used to map the locations of proteins on chromatin. Importantly, LEDGF fusions in which the chromatin binding component of LEDGF is replaced with a module that recognizes specific histone marks direct integration to those marks, confirming integration occurs efficiently on nucleosomes in cells. It is worth noting that it is possible to redirect integration to portions of the host genome that are poorly expressed, which, when taken with the data on integration into LADs (integration in the absence of a CPSF6 interaction) shows that there are circumstances in which there is reasonably efficient integration of HIV DNA in portions of the genome in which there are few if any R-loops.

Although R-loops may not wrap around nucleosomes, long and stable R-loops likely cover stretches of DNA corresponding to multiple nucleosomes (10). For example, R-loops are associated with high levels of histone marks, such as H3K36me3, which LEDGF recognizes (2, 11). R-loops dynamically regulate the chromatin architecture. Possibly by altering nucleosome occupancy, positioning, or turnover, R-loop structures relieve superhelical stress and are often associated with open chromatin marks and active enhancers (2, 10). These features are also distributed over HIV-1 integration sites (12). In the Discussion section of the revised manuscript, we explored the R-loop molding mechanisms in the host genomic environment for HIV-1 integration site selection and its potential collaborative role with LEDGF/p75 and CPSF6 governing HIV-1 integration site selection.

By carefully revising our original manuscript, with respect to the reviewer's comment, we recognized the need to tone down our statements. We found that the title, abstract, and discussion of our original manuscript includes phrases, such as, “HIV-1 targets Rloops for integration,” which may overstate our finding on the role of R-loop in HIV-1 integration site selection. We replaced these phrases. For example, we used phrases, such as “HIV-1 favors vicinity regions of R-loop for the viral genome integration,” in the revised manuscript. We apologize for the inconvenience caused by the unclear and non-specific details of our findings.

(2.3) Given that HIV DNA is known to preferentially integrate into expressed genes and that R-loops must necessarily involve expressed RNA, it is not surprising that there is a correlation between HIV integration and regions of the genome to which R loops have been mapped. However, it is important to remember that correlation does not necessarily imply causation.

We understand the reviewer's concern regarding the possibility of a coincidental correlation between the R-loop regions and HIV-1 integration sites, particularly when the interpretation of this correlation is primarily based on a global analysis.

Therefore, we designed pgR-poor and pgR-rich cell lines, which we believe are suitable models for distinguishing between integration events driven by transcription and the presence of R-loops. Although the two cell lines showed comparable levels of transcription at the designated region upon DOX treatment via TRE promoter activation (Fig. 3B of the revised manuscript), only pgR-rich cells formed R-loops at the designated regions (Fig. 3C of the revised manuscript). When infected with HIV1, pgR-rich cells, but not pgR-poor cells, showed higher infectivity after DOX treatment (Fig. 3D of the revised manuscript). Moreover, we quantified site-specific integration events in the R-loop regions, and found that a greater number of integration events occurred in designated regions of the pgR-rich cellular genome upon R-loop induction by DOX treatment, but not in pgR-poor cells (Fig. 3E of the revised manuscript). Therefore, we concluded that transcriptional activation without an R-loop (in pgR-poor cells) may not be sufficient to drive HIV-1 integration. We believe that our work provides strong evidence for a causative relationship between R-loop formation/Rloop sites and HIV-1 integration. We hope that our explanation addresses your concerns. Thank you.

If we consider some of the problems in the experiments that are described in the manuscript:

(2.4) In an infected individual, cells are almost always infected by a single virion and the infecting virion is not accompanied by large numbers of damaged or defective virions. This is a key consideration: the claim that infection by HIV affects R-loop formation in cells was done with a VSVg vector in experiments in which there appears to have been about 6000 virions per cell. Although most of the virions prepared in vitro are defective in some way, that does not mean that a large fraction of the defective virions cannot fuse with cells. In normal in vivo infections, HIV has evolved in ways that avoid signaling infected the cell of its presence. To cite an example, carrying out reverse transcription in the capsid/core prevents the host cell from detecting (free) viral DNA in the cytoplasm. The fact that the large effect on R-loop formation which the authors report still occurs in infections done in the absence of reverse transcription strengthens the probability that the effects are due to the massive amounts of virions present, and perhaps to the presence of VSVg, which is quite toxic. To have physiological relevance, the infections would need to be carried out with virions that contain HIV even under circumstances in which there is at most one virion per cell.

Our virus production and in vitro and ex vivo HIV-1 infection experimental conditions, designed for infecting cell types, such as HeLa cells and primary CD4+ T cells with VSV-G pseudotyped HIV, were based on a comprehensive review of numerous references. At the very beginning of this study, we tested HIV-1-specific host genomic R-loop induction using empty virion particles (virus-like particles, VLP) or other types of viruses (non-retrovirus, SeV; retroviruses, FMLV and FIV), all produced with a VSV G protein donor. We could not include a control omitting the VSV G protein or using natural HIV-1 envelope protein to prevent viral spread in culture. We observed that despite all types of virus stocks being prepared using VSV-G, only cells infected with HIV-1 viruses showed R-loop signal enrichment (Author response image 3). Therefore, we omitted the control for the VSV G protein in subsequent analyses, such as DRIPcseq. We have also revised our manuscript to provide a clearer description of the experimental conditions. In particular, we now clearly stated that we used VSV-G pseudotyped HIV-1 in this study, throughout the abstract, results, and discussion sections of the revised manuscript. Thank you.

**Author response image 3. sa3fig3:** Dot blot analysis of cellular R-loops in cells infected with different virus particles. (A) Dot blot analysis of the R-loop in gDNA extracts from HIV-1 infected U2OS cells with MOI of 0.6 harvested at 6 hpi. The gDNA extracts were incubated with or without RNase H in vitro before membrane loading (anti-S9.6 signal). (B) Dot blot analysis of the R-loop in gDNA extracts from HeLa cells infected with 0.3 MOI of indicated viruses. The infected cells were harvested at 6 hpi. The gDNA extracts were incubated with or without RNase H in vitro before membrane loading (anti-S9.6 signal).

HIV-1 co-infection may also be expected in cell-free HIV-1 infections. However, it was previously suggested that the average number of infection events varies within 1.02 to 1.65 based on a mathematical model that estimates the frequency of multiple infections with the same virus (Figure 4c of Ito et al., Sci. Rep, 2017; 6559) (13).

(2.5) Using the Sso7d version of HIV IN in the in vitro binding assays raises some questions, but that is not the real question/problem. The real problem is that the important question is not what/how HIV IN protein binds to, but where/how an intasome binds. An intasome is formed from a combination of IN bound to the ends of viral DNA. In the absence of viral DNA ends, IN does not have the same structure/organization as it has in an intasome. Moreover, HIV IN (even Sso7d, which was modified to improve its behavior) is notoriously sticky and hard to work with. If viral DNA had been included in the experiment, intasomes would need to be prepared and purified for a proper binding experiment. To make matters worse, there are multiple forms of multimeric HIV IN and it is not clear how many HIV INs are present in the PICs that actually carry out integration in an infected cell.

As the reviewer has noted, HIV IN, even with Sso7d tagging, is difficult. We attempted the purification of viral DNA (vDNA)-bound PICs using either Sso7d-tagged HIV-1 integrase proteins or non-tagged HIV-1 integrase proteins (F185K/C280S), procured from the NIH HIV reagent program (HRP-20203), following the method described by Passos et al., Science, 2017; 355 (89-92) (3). Despite multiple attempts, we were unable to purify the vDNA-bound IN protein complexes for in vitro assays. However, through multiple biochemical experiments, we believe that we have successfully demonstrated the interaction between cellular R-loops and HIV-1 integrase proteins both in cells and in vitro (Fig. 5A–F of the revised manuscript). We also observed a close association between integrase proteins and host cellular Rloops in HIV-1-infected cells, using a fluorescent recombinant virus (HIV-IN-EGFP) with intact IN-EGFP PICs (Fig. 5G of the revised manuscript).

(2.6) As an extension of comment 2, the proper association of an HIV intasome/PIC with the host genome requires LEDGF and the appropriate nucleic acid targets need to be chromatinized.

The interaction between cellular R-loops and HIV-1 integrase proteins in HeLa cells endogenously expressing LEDGF/p75 was examined using reciprocal immunoprecipitation assays in Fig. 5C–F, S6B, and S6D Fig. of the revised manuscript. In addition, as discussed in more detail in our response to comment [28], we observed a close association between host cellular R-loops and HIV-1 integrase proteins by PLA assay, in HIV-1-infected HeLa cells.

(2.7) Expressing any form of IN, by itself, in cells to look for what IN associates with is not a valid experiment. A major factor that helps to determine both where integration takes place and the sites chosen for integration is the transport of the viral DNA and IN into the nucleus in the capsid core. However, even if we ignore that important part of the problem, the IN that the authors expressed in HeLa cells won't be bound to the viral DNA ends (see comment 2), even if the fusion protein would be able to form an intasome. As such, the IN that is expressed free in cells will not form a proper intasome/PIC and cannot be expected to bind where/how an intasome/PIC would bind.

As discussed in more detail in our response to comment [2-8], we believe that our PLA experiment using the pVpr-IN-EGFP virus, which has previously been examined for virion integrity, as well as the IN-EGFP PICs (14), demonstrated a close association between host cellular R-loops and HIV-1 integrase proteins in HIV-1-infected cells.

(2.8) As in comment 1, for the PLA experiments presented in Figure 5 to work, the number of virions used per cell (which differs from the MOI measured by the number of cells that express a viral marker) must have a high, which is likely to have affected the cells and the results of the experiment. However, there is the additional question of whether the IN-GFP fusion is functional. The fact that the functional intasome is a complex multimer suggests that this could be a problem. There is an additional problem, even if IN-GFP is fully functional. During a normal infection, the capsid core will have delivered copies of IN (and, in the experiments reported here, the IN-GFP fusion) into the nucleus that is not part of the intasome. These "free" copies of IN (here IN-GFP) are not likely to go to the same sites as an intasome, making this experiment problematic (comment 4).

The HIV-IN-EGFP virus stock was produced by polyethylenimine-mediated transfection of HEK293T cells with 6 µg of pVpr-IN-EGFP, 6 µg of HIV-1 NL4-3 noninfectious molecular clone (pD64E; NIH AIDS Reagent Program 10180), and 1 µg of pVSV-G as previously described in (14), and described in the Materials and Methods section of our manuscript. The pVpr-IN-EGFP vector used to produce HIV-1-IN-EGFP virus stock was provided by Anna Cereseto group (Albanese et al., PLOS ONE, 2008; 6(6); Ref 34 of the revised manuscript). It was previously reported that the HIV-1INEGFP virions produced by IN-EGFP trans-incorporation through Vpr are intact and infective viral particles (Figure 1 of Albanese et al., PLOS ONE, 2008; 6(6)). Therefore, we believe that the HIV-IN-EGFP used in our PLA experiments was functional.

Additionally, Albanese et al. showed that the EGFP signal of HIV-IN-EGFP virions colocalizes with the viral protein matrix (p17MA) and capsid (P24CA) as well as with the newly synthesized cDNA produced by reverse transcriptase by labeling and visualizing the synthesized cDNA (14). In addition, the fluorescent recombinant virus (HIV-INEGFP) was structurally intact at the nuclear level (Figure 6 of Albanese et al., PLOS ONE, 2008; 6(6)). Therefore, we believe that our PLA experimental result is not likely misled as the reviewer concerns due to the integrity of the HIV-IN-EGFP virion as well as IN-EGFP PICs.

Furthermore, the in vitro HIV-1 infection setting of our PLA experiments was carefully determined based on multiple studies that performed image-based assays on HIV-1infected cells. For instance, Albanese et al. infected 4 × 104 cells with viral loads equivalent to 1.5 or 3 µg of HIV-1 p24 for their immunofluorescence analysis, in their previous report (14). We titrated the fluorescent HIV-1 virus stocks by examining both the multiplicity of infection (MOI) and quantifying the HIV-1 p24 antigen content (Author response image 4). In our calculation, we infected 5 × 104 HeLa cells with viral loads equivalent to 1.3 ug of HIV-1 p24, which is indicated as 2 MOI in Fig. 5G of our manuscript, for our PLA experiments.

Image-Based Assays often require increased and enhanced signal for statistical robustness. For example, Achuthan et al. infected cells with VSV-G-pseudotyped HIV1 at the approximate MOI of 350 for vDNA and PIC visualization (15). Therefore, we believe our experimental condition for PLA experiments, which we carefully designed based on previous study that are frequently referred, are reasonable. We really hope that our discussion sufficiently addressed the reviewer’s concern.

**Author response image 4. sa3fig4:** Gating strategy used to determine HIV-1-infectivity in HeLa cells at 48 hpi. Cells were infected with a known p24 antigen content in the stock of the VSV-G-pseudotyped HIV-1-EGFP-virus. The percentages of GFP-positive cell population are indicated.

(2.9) In the Introduction, the authors state that the site of integration affects the probability that the resulting provirus will be expressed. Although this idea is widely believed in the field, the actual data supporting it are, at best, weak. See, for example, the data from the Bushman lab showing that the distribution of integration sites is the same in cells in which the integrated proviruses are, and are not, expressed. However, given what the authors claim in the introduction, they should be more careful in interpreting enzyme expression levels (luciferase) as a measure of integration efficiency in experiments in which they claim proviruses are integrated in different places.

We thank the reviewer for the constructive comment. We have changed the statement in Lines 41–42 in the Introduction section of our original manuscript to “The chromosomal landscape of HIV-1 integration influences proviral gene expression, persistence of integrated proviruses, and prognosis of antiretroviral therapy.” (Lines 39-41 of the revised manuscript). We believe that this change can tone-down the relevance between the site of integration and the provirus expression level.

The piggyBac transposase randomly insert the “cargo (transposon)” into TTAA chromosomal sites of the target genome, generating efficient insertions at different genomic loci (16, 17). We believe that this random insertion of the pgR-poor/rich vector mediated by the piggyBac system allows us not to mislead the R-loop-mediated HIV1 integration site because of the genome locus bias of the vector insertion. Therefore, Figure 3 in our manuscript does not claim any relevance between the site of integration and the resulting provirus expression levels. Instead, as noted in Line 214 of the revised manuscript, using the luciferase reporter HIV-1 virus, we attempted to examine HIV-1 infection in cells with an "extra number of R-loops” in the host cellular genome. We observed that pgR-rich cells showed higher luciferase activity upon DOX treatment than pgR-poor cells (Fig. 3D of the revised manuscript). We believe that this is because a greater number of HIV-1 integration events may occur in pgR-rich cells, where DOX-inducible de novo R-loop regions are introduced. This has been further examined in Fig. 3E–G of the revised manuscript. We hope this explanation clarifies the Figure 3. Thank you.

(2.10) Using restriction enzymes to create an integration site library introduces biases that derive from the uneven distribution of the recognition sites for the restriction enzymes.

As described in the Materials and Methods section, we adopted a sequencing library construction method using a previously established protocol (18, 19). Although we recognize the advantages of DNA fragmentation by sonication, in in vitro or ex vivo HIV-1 infection settings, where the multiplicity of infection is carefully determined based on multiple references, more copies of integrated viral sequences are expected compared to that in samples from infected patients (18). Therefore, in these settings, restriction enzyme-based DNA fragmentation and ligation-mediated PCR sequencing are well-established methods that provide significant data sources for HIV-1 integration site sequencing (15, 20-22). Furthermore, our data showing the proportion of integration sites over R-loop regions (Fig. 4B of the revised manuscript) are presented alongside the respective random controls (i.e., proportion of integration sites within the 30-kb windows centered on randomized DRIPc-seq peaks, gray dotted lines; control comparisons between randomized integration sites with DRIPc-seq peaks, black dotted lines; and randomized integration sites with randomized DRIPcseq peaks, gray solid lines), which do not show such a correlation between the HIV-1 integration sites and nearby areas of the R-loop regions. Therefore, we believe that our results from the integration site sequencing data analysis are unlikely to be biased.

**Reviewer #3 (Public Review):**
In this manuscript, Park and colleagues describe a series of experiments that investigate the role of R-loops in HIV-1 genome integration. The authors show that during HIV-1 infection, R-loops levels on the host genome accumulate. Using a synthetic R-loop prone gene construct, they show that HIV-1 integration sites target sites with high R-loop levels. They further show that integration sites on the endogenous host genome are correlated with sites prone to R-loops. Using biochemical approaches, as well as in vivo co-IP and proximity ligation experiments, the authors show that HIV-1 integrase physically interacts with R-loop structures.My primary concern with the paper is with the interpretations the authors make about their genome-wide analyses. I think that including some additional analyses of the genome-wide data, as well as some textual changes can help make these interpretations more congruent with what the data demonstrate. Here are a few specific comments and questions:

We are grateful for the time and effort we spent on our behalf and the reviewer’s appreciation for the novelty of our work, in particular, R-loop induction by HIV-1 infection and the correlation between host R-loops and the genomic site of HIV-1 integration. In the following sections, we provide our responses to your comments and suggestions. Your comments are in italics. We have carefully addressed the following issues.

(3.1) I think Figure 1 makes a good case for the conclusion that R-loops are more easily detected HIV-1 infected cells by multiple approaches (all using the S9.6 antibody). The authors show that their signals are RNase H sensitive, which is a critical control. For the DRIPc-Seq, I think including an analysis of biological replicates would greatly strengthen the manuscript. The authors state in the methods that the DRIPc pulldown experiments were done in biological replicates for each condition. Are the increases in DRIPc peaks similar across biological replicates? Are genomic locations of HIV-1-dependent peaks similar across biological replicates? Measuring and reporting the biological variation between replicate experiments is crucial for making conclusions about increases in R-loop peak frequency. This is partially alleviated by the locus-specific data in Figure S3A. However, a better understanding of how the genome-wide data varies across biological replicates will greatly enhance the quality of Figure 1.

DRIPc-seq experiments were conducted with two biological replicates. To define consensus DRIPc-seq peaks using these two replicates, we used two methods applicable to ChIP-seq analysis: the irreproducible discovery rate (IDR) method and sequencing data pooling. We found that the sequencing data pooling method yielded significantly more DRIPc-seq peaks than consensus peak identification through IDR, and we decided to utilize R-loop peaks from pooled sequencing data for our downstream analyses, as described in the figure legends and Materials and Methods of the revised manuscript.

As noted by the reviewer, it is important to verify whether the increasing trend in the number of R-loop peaks and genomic locations of HIV-1 dependent R-loops were consistently observed across the two biological replicates. Therefore, we independently performed R-loop calling on each replicate of the sequencing data of primary CD4+ T cells from two individual donors to verify that the increase in R-loop numbers was consistent (Author response image 5). Additionally, the overlap of the R-loop peaks between the two replicates was statistically significant across the genome (Author response table 1). Thank you.

**Author response image 5. sa3fig5:** Bar graph indicating DRIPc-seq peak counts for HIV-1-infected primary CD4+ T cells harvested at the indicated hours post infection (hpi). Pre-immunoprecipitated samples were untreated (−) or treated (+) with RNase H, as indicated. Each dot corresponds to an individual data set from two biologically independent experiments.

**Author response table 1. sa3table1:** DRIPc-seq peak length and Chi-square p-value in CD4+ T cells from individual donor 1 and 2.

	Total DRIPc-seq peak length(replicate 1, bp)	Total DRIPc-seq peak length(replicate 2, bp)	Total DRIPc-seq peak length(overalp, bp)	Chi-squarep-value
Ohpi	1910609	4722514	540640	p < 1e-308
3hpi	3630859	3308270	835457	p < 1e-308
6hpi	4195887	8496266	1485568	p < 1e-308
12hpi	4323824	13714402	1903473	p < 1e-308

(3.2) I think that the conclusion that R-loops "accumulate" in infected cells is acceptable, given the data presented. However, in line 134 the authors state that "HIV1 infection induced host genomic R-loop formation". I suggest being very specific about the observation. Accumulation can happen by (a) inducing a higher frequency of the occurrence of individual R-loops and/or (b) stabilizing existing R-loops. I'm not convinced the authors present enough evidence to claim one over the other. It is altogether possible that HIV-1 infection stabilizes R-loops such that they are more persistent (perhaps by interactions with integrase?), and therefore more easily detected. I think rephrasing the conclusions to include this possibility would alleviate my concerns.

We thank the reviewer for the considerable discussion on our manuscript. We have now changed Line 134 to, “HIV-1 infection induces host genomic R-loop enrichment” (Lines 132-133 of the revised manuscript), and added a new conclusion sentence implicating the possible explanation for the R-loop signal enrichment upon HIV-1 infection (Lines 133–135 of the revised manuscript), according to the reviewer's suggestion.

(3.3) A technical problem with using the S9.6 antibody for the detection of R-loops via microscopy is that it cross-reacts with double-stranded RNA. This has been addressed by the work of Chedin and colleagues (as well as others). It is absolutely essential to treat these samples with an RNA:RNA hybrid-specific RNase, which the authors did not include, as far as their methods section states. Therefore, it is difficult to interpret all of the immunofluorescence experiments that depend on S9.6 binding.

We understand the reviewer's concern regarding the cross-reactivity of the S9.6 antibody with more abundant dsRNA, particularly in imaging applications. We carefully designed the experimental and analytical methods for R-loop detection using microscopy. For example, we pre-extracted the cytoplasmic fraction before staining with the S9.6 antibody and quantified the R-loop signal by subtracting the nucleolar signal. Both of these steps were taken to eliminate the possibility of misdetecting Rloops via microscopy because of the prominent cytoplasmic and nucleolar S9.6 signals, which primarily originate from ribosomal RNA. In addition, we included R-loop negative control samples in our microscopy analysis that were subjected to intensive RNase H treatment (60U/mL RNase H for 36 h) and observed a significant reduction in the S9.6 signal (Figure 1E of the revised manuscript). RNase H-treated samples served as essential and widely accepted negative controls for R-loop detection.

We would like to point out that recent studies have reported strong intrinsic specificity of S9.6 anybody for DNA:RNA hybrid duplex over dsDNA and dsRNA, along with the structural elucidations of S9.6 antibody recognition of hybrids (23, 24). Therefore, our interpretation of host cellular R-loop enrichment after HIV-1 infection using S9.6 antibodies in multiple biochemical approaches is well supported. Nevertheless, we agree with the reviewer's opinion that additional negative controls for the detection of R-loops via microscopy, such as RNase T1-and RNase III-treated samples, could improve the robustness and accuracy of R-loop imaging data (25).

(3.4) Given that there is no clear correlation between expression levels and R-loop peak detection, combined with the data that show increased detection of R-loop frequency in non-genic regions, I think it will be important to show that the R-loop forming regions are indeed transcribed above background levels. This will help alleviate possible concerns that there are technical errors in R-loop peak detection.

Figures S5D and S5E in the revised manuscript show the relative gene expression levels of the R-loop-forming positive regions (P1-3) and the referenced Rloop-positive loci (RPL13A and CALM3). The gene expression levels of these R-loopforming regions were significantly higher than those of the ECFP or mAIRN genes without DOX treatment, which can be considered background levels of transcription in cells. Thank you.

(3.5) In Figures 4C and D the hashed lines are not defined. It is also interesting that the integration sites do not line up with R-loop peaks. This does not necessarily directly refute the conclusions (especially given the scale of the genomic region displayed), but should be addressed in the manuscript. Additionally, it would greatly improve Figure 4 to have some idea about the biological variation across replicates of the data presented 4A.

We thank the reviewer for the considerable comment on our study. First of all, we added an annotation for the dashed lines in the figure legends of Figures 4C and 4D in the revised manuscript.

We agree with the reviewer's interpretation of the relationship between the integration sites and R-loop peaks. Primarily based on our current data, we believe R-loop structures are bound by HIV-1 integrase proteins and lead HIV-1 viral genome integration into the “vicinity” regions of the host genomic R-loops. We displayed a large-scale genomic region (30-kb windows) to present integration sites surrounding R-loop centers because an R-loop can be multi-kilobase in size (1, 2). Depending on the immunoprecipitation and library construction methods, the R-loop peaks varied in size, and the peak length showed a wide distribution (Figure 3B of Malig et al., 2020, Figure 1B of Sanz et al., 2016, and Figure 2A of the revised manuscript). Therefore, presenting integration site events within a wide window of R-loop peaks could be more informative and better reflect the current understanding of R-loop biology.

R-loop formation recruits diverse chromatin-binding protein factors, such as H3K4me1, p300, CTCF, RAD21, and ZNF143 (Figure 6A and 6B of Sanz et al., 2016) (26), which allow R-loops to exhibit enhancer and insulator chromatin states, which can act as distal regulatory elements (26, 27). We have demonstrated physical interactions between host cellular R-loops and HIV-1 integrase proteins (Figure 5 of the revised manuscript), therefore, we believe that this ‘distal regulatory element-like feature’ of the R-loop can be a potential explanation for how R-loops drive integration over longrange genomic regions.

According to your suggestion, we added this explanation to the relevant literature in the Discussion section of the revised manuscript.

Author response image 6 which represents the biological variation across replicates of the data shown in Figure 4A. The integration site sequencing data for Jurkat cells were adopted from SRR12322252 (4), which consists of the integration site sequencing data of HIV-1-infected wild type Jurkat cells with one biological replicate. We hope that our explanations and discussion have successfully addressed your concerns. Thank you.

**Author response image 6. sa3fig6:** Bar graphs showing the quantified number of HIV-1 integration sites per Mb pair in total regions of 30-kb windows centered on DRIPc-seq peaks from HIV-1 infected HeLa cells and primary CD4+ T cells (magenta) or non-R-loop region in the cellular genome (gray). Each dot corresponds to an individual data set from two biologically independent experiments.

(3.6) The authors do not adequately describe the Integrase mutant that they use in their biochemical experiments in Figure 5A. Could this impact the activity of the protein in such a way that interferes with the interpretation of the experiment? The mutant is not used in subsequent experiments for Figure 5 and so even though the data are consistent with each other (and the conclusion that Integrase interacts with R-loops) a more thorough explanation of why that mutant was used and how it impacts the biochemical activity of the protein will help the interpretation of the data presented in Figure 5.

We appreciate the reviewer’s suggestions. In our EMSA analysis, we purified and used Sso7d-tagged HIV-1 integrase proteins with an active-site amino acid substitution, E152Q. First, we used the Sso7d-tagged HIV-1 integrase protein, as it has been suggested in previous studies that the fusion of small domains, such as Sso7d (DNA binding domain) can significantly improve the solubility of HIV integrase proteins without affecting their ability to assemble with substrate nucleic acids and their enzymatic activity Figure 1B of Li et al., PLOS ONE, 2014;9 (8) (28, 29). We used an integrase protein with an active site amino acid substitution, E152Q, in our mobility shift assay, because the primary goal of this experiment was to examine the ability of the protein to bind or form a complex with different nucleic acid substrates. We thought that abolishing the enzymatic activity of the integrase protein, such as 3'-processing that cleaves DNA substrates, would be more appropriate for our experimental objective. This Sso7d tagged- HIV-1 integrase with the E152Q mutation has also been used to elucidate the structural model of the integrase complex with a nucleic acid substrate by cryo-EM (3) and has been shown to not disturb substrate binding.

Based on the reviewer’s comments, we have added a description of the E152Q mutant integrase protein in Lines 268–270 of the revised manuscript. Thank you.

**Reviewer #3 (Recommendations For The Authors):**
The paper suffers from many grammatical errors, which sometimes interfere with the interpretations of the experiments. In the view of this reviewer, the manuscript must be carefully revised prior to publication. For example, lines 247-248 "Intasomes consist of HIV-1 viral cDNA and HIV-1 coding protein, integrases." It is unclear from this sentence whether there are multiple integrases or multiple proteins that interact with the viral genome to facilitate integration. This makes the subsequent experiments in Figure 5 difficult to interpret. There are many other examples, too numerous to point out individually.

We thoughtfully revised the original manuscript, making the best efforts to provide clearer details of our findings. We believe that we have made substantial changes to the manuscript, including Lines 247–248 of the original manuscript that the reviewer noted. Furthermore, the revised manuscript was edited by a professional editing service. Thank you.

(1) M. Malig, S. R. Hartono, J. M. Giafaglione, L. A. Sanz, F. Chedin, Ultra-deep Coverage Singlemolecule R-loop Footprinting Reveals Principles of R-loop Formation. J Mol Biol 432, 22712288 (2020).

(2) L. A. Sanz et al., Prevalent, Dynamic, and Conserved R-Loop Structures Associate with Specific Epigenomic Signatures in Mammals. Mol Cell 63, 167-178 (2016).

(3) D. O. Passos et al., Cryo-EM structures and atomic model of the HIV-1 strand transfer complex intasome. Science 355, 89-92 (2017).

(4) W. Li et al., CPSF6-Dependent Targeting of Speckle-Associated Domains Distinguishes Primate from Nonprimate Lentiviral Integration. mBio 11, (2020).

(5) P. A. Ginno, Y. W. Lim, P. L. Lott, I. Korf, F. Chedin, GC skew at the 5' and 3' ends of human genes links R-loop formation to epigenetic regulation and transcription termination. Genome Res 23, 1590-1600 (2013).

(6) S. Hamperl, M. J. Bocek, J. C. Saldivar, T. Swigut, K. A. Cimprich, Transcription-Replication Conflict Orientation Modulates R-Loop Levels and Activates Distinct DNA Damage Responses. Cell 170, 774-786 e719 (2017).

(7) H. O. Ajoge et al., G-Quadruplex DNA and Other Non-Canonical B-Form DNA Motifs Influence Productive and Latent HIV-1 Integration and Reactivation Potential. Viruses 14, (2022).

(8) I. K. Jozwik et al., B-to-A transition in target DNA during retroviral integration. Nucleic Acids Res 50, 8898-8918 (2022).

(9) F. Chedin, C. J. Benham, Emerging roles for R-loop structures in the management of topological stress. J Biol Chem 295, 4684-4695 (2020).

(10) F. Chedin, Nascent Connections: R-Loops and Chromatin Patterning. Trends Genet 32, 828838 (2016).

(11) P. B. Chen, H. V. Chen, D. Acharya, O. J. Rando, T. G. Fazzio, R loops regulate promoterproximal chromatin architecture and cellular differentiation. Nat Struct Mol Biol 22, 9991007 (2015).

(12) A. R. Schroder et al., HIV-1 integration in the human genome favors active genes and local hotspots. Cell 110, 521-529 (2002).

(13) Y. Ito et al., Number of infection events per cell during HIV-1 cell-free infection. Sci Rep 7, 6559 (2017).

(14) A. Albanese, D. Arosio, M. Terreni, A. Cereseto, HIV-1 pre-integration complexes selectively target decondensed chromatin in the nuclear periphery. PLoS One 3, e2413 (2008).

(15) V. Achuthan et al., Capsid-CPSF6 Interaction Licenses Nuclear HIV-1 Trafficking to Sites of Viral DNA Integration. Cell Host Microbe 24, 392-404 e398 (2018).

(16) X. Li et al., piggyBac transposase tools for genome engineering. Proc Natl Acad Sci U S A 110, E2279-2287 (2013).

(17) Y. Cao et al., Identification of piggyBac-mediated insertions in Plasmodium berghei by next generation sequencing. Malar J 12, 287 (2013).

(18) E. Serrao, P. Cherepanov, A. N. Engelman, Amplification, Next-generation Sequencing, and Genomic DNA Mapping of Retroviral Integration Sites. J Vis Exp, (2016).

(19) K. A. Matreyek et al., Host and viral determinants for MxB restriction of HIV-1 infection. Retrovirology 11, 90 (2014).

(20) G. A. Sowd et al., A critical role for alternative polyadenylation factor CPSF6 in targeting HIV-1 integration to transcriptionally active chromatin. Proc Natl Acad Sci U S A 113, E10541063 (2016).

(21) B. Lucic et al., Spatially clustered loci with multiple enhancers are frequent targets of HIV-1 integration. Nat Commun 10, 4059 (2019).

(22) P. K. Singh, G. J. Bedwell, A. N. Engelman, Spatial and Genomic Correlates of HIV-1 Integration Site Targeting. Cells 11, (2022).

(23) C. Bou-Nader, A. Bothra, D. N. Garboczi, S. H. Leppla, J. Zhang, Structural basis of R-loop recognition by the S9.6 monoclonal antibody. Nat Commun 13, 1641 (2022).

(24) Q. Li et al., Cryo-EM structure of R-loop monoclonal antibody S9.6 in recognizing RNA:DNA hybrids. J Genet Genomics 49, 677-680 (2022).

(25) J. A. Smolka, L. A. Sanz, S. R. Hartono, F. Chedin, Recognition of RNA by the S9.6 antibody creates pervasive artifacts when imaging RNA:DNA hybrids. J Cell Biol 220, (2021).

(26) L. A. Sanz, F. Chedin, High-resolution, strand-specific R-loop mapping via S9.6-based DNARNA immunoprecipitation and high-throughput sequencing. Nat Protoc 14, 1734-1755 (2019).

(27) M. Merkenschlager, D. T. Odom, CTCF and cohesin: linking gene regulatory elements with their targets. Cell 152, 1285-1297 (2013).

(28) M. Li, K. A. Jurado, S. Lin, A. Engelman, R. Craigie, Engineered hyperactive integrase for concerted HIV-1 DNA integration. PLoS One 9, e105078 (2014).

(29) M. Li et al., A Peptide Derived from Lens Epithelium-Derived Growth Factor Stimulates HIV1 DNA Integration and Facilitates Intasome Structural Studies. J Mol Biol 432, 2055-2066 (2020).